# MECHANISTIC UNLEARNING:
# ROBUST KNOWLEDGE UNLEARNING AND EDITING VIA MECHANISTIC LOCALIZATION

## ABSTRACT

Methods for knowledge editing and unlearning in large language models seek to edit or remove undesirable knowledge or capabilities without compromising general language modeling performance. This work investigates how mechanistic interpretability—which, in part, aims to identify model components (circuits) associated to specific interpretable mechanisms that make up a model capability—can improve the precision and effectiveness of editing and unlearning. We find a stark difference in unlearning and edit robustness when training components localized by different methods. We highlight an important distinction between methods that localize components based primarily on preserving outputs, and those finding high level mechanisms with predictable intermediate states. In particular, localizing edits/unlearning to components associated with the *lookup-table mechanism* for factual recall 1) leads to more robust edits/unlearning across different input/output formats, and 2) resists attempts to relearn the unwanted information, while also reducing unintended side effects compared to baselines, on both a sports facts dataset and the CounterFact dataset across multiple models. We also find that certain localized edits disrupt the latent knowledge in the model more than any other baselines, making unlearning more robust to various attacks.

## 1 INTRODUCTION

Large language models (LLMs) often learn to encode undesirable knowledge. The possibility of selectively editing or unlearning this type of knowledge is viewed as paramount for ensuring accuracy, fairness, and control of AI. Yet, editing and unlearning of knowledge from these models remains challenging.

Common editing and unlearning methods often come at the cost of affecting other general or tangential knowledge or capabilities within the model. Moreover, the edits achieved through these methods may not be robust – e.g., slight variations in the prompt formulation can often still elicit the original fact or capability, or the original answers are still present/extractable given white-box access.

Some recent work has explored editing or unlearning techniques that rely on mechanistic interpretability methods attempting to trace which components of a network store specific facts (Meng et al., 2023). These methods, such as causal tracing or attribution patching, focus on measuring how output or task accuracy is affected when clean/corrupted input is patched into specific components.

We coin a new term to categorize localizations which measure causal effects of components on only the output: *Output-Tracing* localizations. The effectiveness of output-tracing (OT) techniques like Causal Tracing for editing has been questioned by Hase et al. (2023). Our research confirms these doubts, finding that localized editing and unlearning of facts based on several existing OT methods often perform equal to or worse than simply updating the entire model. This is particularly evident when evaluating the robustness of edits against prompt variations and relearning, and when probing for remaining latent knowledge.

Another style of interpretability techniques first breaks down computations into high-level mechanisms with predictable intermediate states. Based on such work by Nanda et al. (2023); Geva et al. (2023), we link certain MLP layers to a fact lookup (FLU) mechanism for facts used in our analysis,

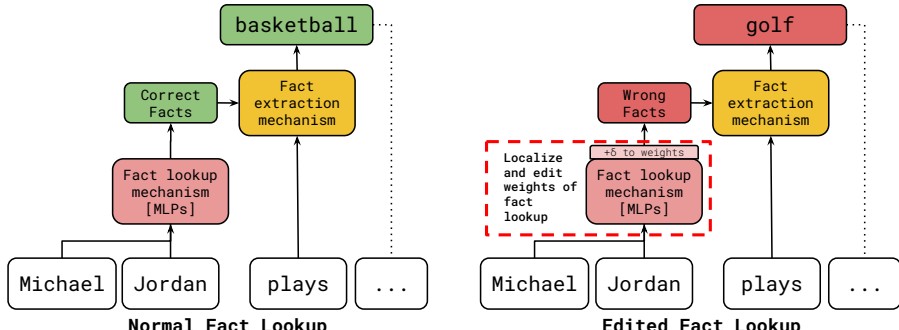

Figure 1: High level depiction of *mechanistic unlearning*. We localize components responsible for fact extraction/enrichment and modify their weights to change the associations, in order to target internal latent representations rather than targeting the output. Graph inspired by Nanda et al. (2023).

that enrich the latent stream with subject attributes but don't directly write to the output. For unlearning and edits of these facts, we only modify components that implement the FLU mechanism. More broadly, we refer to editing and unlearning that acts on components of the model identified by mechanistic intermediate component analysis as *mechanistic unlearning*. We demonstrate that FLU *mechanistic unlearning* leads to better trade-offs between edits/unlearning and maintaining performance on general language modelling capabilities, compared to edits done using OT or without any localization. Further, it exhibits improved robustness to re-learning and alternative prompting, and we demonstrate that the latent knowledge is also perturbed.

**Summary of Contributions**

- We perform a rigorous evaluation of several standard editing approaches on factual recall tasks, and we identify mechanisms for factual lookup and attribute extraction on Gemma-7B, Gemma-2-9B, and Llama-3-8b. We demonstrate that gradient-based editing localized on the factual lookup mechanism is more robust than OT localizations and baselines across multiple datasets, models, and evaluations.

- We demonstrate that it is more difficult to elicit the forgotten ground truth answers using alternative prompting with FLU localizations. We also demonstrate slower or no relearning of the ground truth answers, retraining edited models on half of the edited set and evaluating them on the other half of the edited set.

- We analyze intermediate representations using probing, and provide further evidence that editing with FLU localization modifies the internal latent information to reflect the desired edited answer more than other localizations and baselines. We also analyze the weights that are modified for each localization, and find that OT techniques and baselines modify the attribute extraction mechanisms more than the fact lookup mechanism.

- We show that editing and unlearning localized on these mechanisms is more parameter efficient, by controlling for the sizes of edits made to the model with weight masking.

## 1.1 RELATED WORK

**Mechanistic Interpretability** is a subfield of AI interpretability, aiming to understand the internal processes of AI models by attributing them to subnetworks (called circuits) within the model (Olah et al., 2020). We focus on the factual recall interpretability literature (Nanda et al., 2023; Geva et al., 2023; Chughtai et al., 2024; Yu et al., 2023), which studies methods that aim to discover mechanisms for the retrieval and formatted extraction of factual information.

**Output tracing** methods aim to automatically find causally important subnetworks of components for a task. Causal Tracing (Meng et al., 2023) and Automated circuit discovery (ACDC) (Conmy et al., 2023) utilize repeated activation patching to attempt to find the subnetworks that are most critical for the model's output on that task. Efficient methods such as attribution patching (Nanda, 2023)

and edge attribution patching (Syed et al., 2023) are linear approximations of activation patching for discovering important components quickly.

**Fact Editing and Machine Unlearning** seek to modify pre-trained models to eliminate or alter learned knowledge such as capabilities or facts. Some prior approaches focus on identifying and removing specific individual training data points, aiming to obtain a model that is "similar" to one that had never trained on these data points (Cao & Yang, 2015; Xu et al., 2023). One formalization of unlearning to match a retrained-from-scratch model is due to Ginart et al. (2019), and is closely inspired by differential privacy (Dwork et al., 2014).

Fact editing focuses on overwriting factual information while preserving overall language generation ability. Meng et al. (2023) attempts to identify MLP modules that are most responsible for factual predictions via Causal Tracing and then applies a rank-one transformation upon these modules to replace factual associations.

In the context of LLMs and safety, techniques such as Helpful-Harmless RLHF (Bai et al., 2022) and Representation Misdirection for Unlearning (Li et al., 2024b) aim to suppress dangerous knowledge or harmful tendencies in LLMs. Li et al. (2024a); Zou et al. (2023; 2024) approach unlearning and dangerous knowledge suppression from a top-down feature view, reading or suppressing linear features related to memorized, harmful, and undesired concepts. A related line of work on safety proposes methods making it difficult to modify open models for use on harmful domains (Tamirisa et al., 2024; Deng et al., 2024; Henderson et al., 2023), including through adversarial relearning.

**Failures of Unlearning and Editing** have been shown for both localized and nonlocalized methods. Patil et al. (2023) extract correct answers to edited facts from the intermediate residual stream and through prompt rephrasing. Yong et al. (2024) show that low-resource languages jailbreak models output unsafe content, and Lo et al. (2024); Lermen et al. (2023); Deeb & Roger (2024) demonstrate that relearning with a small amount of compute/data causes models to regain undesirable knowledge/tendencies. Even without explicit finetuning, Xhonneux et al. (2024) show that in-context learning alone suffices to reintroduce undesirable knowledge despite the model being designed to refuse to output such knowledge. Lee et al. (2024) shows that even after alignment techniques are applied to make models nontoxic, toxicity representations are still present, just not triggered - they argue that this is a reason that models lack robustness and can still be jailbroken to trigger this unwanted behavior.

Hong et al. (2024) evaluates unlearning by measuring residual knowledge left in internal activations, and demonstrate that current approaches fail to remove this residual knowledge and thus can be exploited. They attempt unlearning by targeting the MLPs these residual knowledge traces reside in, but fail to find a non-oracle unlearning approach that successfully removes residual information.

## 2 METHODS

Our experiments are designed to test the effectiveness of localization for editing of facts. In this section we describe the tasks used and the localization and editing methods evaluated.

### 2.1 EDITING TASKS

We focus on editing subsets of two datasets: (1) Sports Facts dataset from Nanda et al. (2023), which contains subject-sport relations across three sports categories for 1567 athletes, and (2) the CounterFact dataset from Meng et al. (2023).

**Sports Facts: Sports-Athlete-Editing, Full-Sports-Editing, and Sports-Unlearning tasks** In the Sports Facts dataset, we edit two general groups of factual associations. For the first editing task, we edit factual associations for a constant set of randomly selected athletes belonging to any of the three sport categories. We test editing these sets of associations by replacing their correct sports with one of the other two incorrect sports (with equal probability). To increase the comprehensiveness of our evaluation, we run experiments with different *forget set* sizes: 16 athletes and 64 athletes. We refer to this task as **Sports-Athlete-Editing**. These chosen forget sets are constant between all localizations. For the second editing task, we unlearn all athlete-sport associations for athletes who play one sport. In this case, we establish a forget set consisting of all the athletes who play one sport

(basketball, baseball, or football), and we edit the association by replacing the athlete's correct sport with golf. For comprehensiveness, we vary the sport that the forget set is constructed from. We refer to this task as **Full-Sports-Editing**. Finally, we also design an unlearning task, **Sports-Unlearning**, where the goal is to unlearn factual associations for all athletes in one of the sports.

**CounterFact-Editing and Sequential-CounterFact-Editing task**   In the CounterFact dataset, following Geva et al. (2023), we first filter the dataset for facts which our models assign higher than 50% probability to the right answer, which varies per model. The goal of our **CounterFact-Editing** task is to edit a constant set of facts, replacing the correct answers with an alternative false target, with the retain set being the rest of the non-forget facts. We vary forget set sizes to be of 16 and 64 facts. In **Sequential-CounterFact-Editing** task, we edit a total of 64 facts by sequentially editing four randomly selected subsets of 16 facts. We test sequential editing here because facts from CounterFact can reside in different parts of the model, so we wish to test if we can exploit different localizations for different facts. These chosen forget sets are constant between all localizations.

**Models**   We implement editing on the Gemma-7B LLM, the Gemma-2-9B LLM, and the Llama-3-8b LLM. We don't use the Pythia-2.8B (Biderman et al., 2023) and GPT-2 models tested in the previous fact interpretability literature because our larger models have stronger general capabilities which we can measure for side effects, and also because our larger models can provide factual knowledge in more input/output formats for more robustness evaluations.

## 2.2 Localization Methods and Baselines

Given a model $M : X \mapsto L$ mapping sequence of tokens $X$ to logits $L \in \mathbb{R}^{|V|}$ over vocabulary $V$, we consider $M$ to be a directed acyclic graph $(C, E)$ with $C$ being a set of model components and $E$ being edges between components. Adopting notation from Elhage et al. (2021), we consider the query, key, value, and output weights of each head along with the input, gate, and output projection weights of each MLP as components.

We are interested in finding $S : C \to \mathbb{R}$, a mapping of components to their importance in a given task. A localization is a set of components $C_\tau := \{c : c \in C, |S(c)| > \tau\}$, where $\tau$ is a threshold. In practice, we fix $\tau$ such that $C_\tau$ contains the same number of parameters in OT, FLU, and random localizations. We use these efficient localization methods for finding these mappings:

**Output Tracing (OT) localization: Causal Tracing and Attribution Patching**   First, we test Causal Tracing, a method for finding components with high direct causal importance for factual associations (Meng et al., 2023). We also use Attribution Patching (Nanda, 2023) as a fast and acceptably accurate approximation of causal tracing to automatically localize over components with high direct and indirect importance. We additionally consider the versions of these localizations with only MLPs (*Causal Tracing MLPs* and *Attribution Patching MLPs*).

We hypothesize that these output-based techniques will prioritize the shared extraction components and other mechanisms for reformatting predictions over the more diffuse FLU components, and thus appear more precise yet leave the underlying latent information present in the model. This might decrease robustness under alternative extraction methods, thus motivating non-OT-based localization, described next. We discuss the precise components/mechanisms highlighted by OT localizations in Appendix A.3.

**Fact Lookup (FLU) localization:**   Next, we use manually derived localizations for MLP layers. For Sports Facts, our localization is inspired by Nanda et al. (2023), who discovered components in Pythia 2.8B responsible for *token concatenation, fact lookup*, and *attribute extraction*. They, along with Geva et al. (2023), find that the fact-lookup stage enriches the latent stream with information about the subject (athlete) at the subject's token position, and the attribute extraction stage extracts the latent sport information and formats it in the final token position. We replicate a key result of their work in our three models by using a linear probing technique and attention ablations to discover which MLPs are sufficient for developing accurate representations of which of the three sports each athlete plays.

For CounterFact, we replicate findings from Geva et al. (2023) and do further causal analyses to identify particular MLPs. We first use path patching from Goldowsky-Dill et al. (2023) to isolate

causally important attribute extraction mechanisms, both attention heads and MLPs, in later layers of the model. Then, we use the same technique to patch between MLPs and the attribution extraction mechanism to find which MLPs enrich the latent stream with information to be extracted. We pick the MLPs that have the largest effect on the logit difference as mediated through this extraction mechanism. More details about the manual analysis for both datasets is outlined in Appendix A.2.1.

Importantly, FLU differs from OT techniques because we consider the causal effects of ablations upon intermediate representations used by the factual recall mechanism, not just the effects on the output. We hypothesize that the optimal location for robust editing is in the fact lookup stage rather than in the attribute extraction stage, because adversaries can develop alternative methods for extracting knowledge from the latent stream through alternative prompts or white-box methods so we want to prevent the knowledge from ever being added to the latent stream. Thus, we exclusively modify the fact lookup MLPs.

**Baselines: Random-MLPs, Random, All-MLPs, and Nonlocalized**   We additionally consider four baselines: one corresponding to $C_\tau = C$ (i.e., no localization, optimizing all the components of the model), another that randomly chooses components, another that trains all MLP components, and another trains a random selection of MLP components. We test the last MLP baselines to determine if our mechanistically localized MLPs are uniquely important - we want to know if the same unlearning performance can be achieved with just the heuristic that training only MLPs improves robustness, or if mechanistic understanding of the role of the component is crucial.

In Appendix A.5, we analyze the proportions of each mechanism (the extraction heads, extraction MLPs, and fact lookup MLPs, by parameter count) that are present in each localization.

The main text focuses on comparing FLU, Causal Tracing, and Nonlocalized, while the appendix has the same figures with all above localizations included, with the same conclusions in every case.

### 2.3 PARAMETER UPDATE METHOD

Once we have a localization $C_\tau$, we run one of the unlearning or editing methods, restricting weight updates to only components in $C_\tau$. We update weights using gradient descent on a combination of loss functions.

**Localized Fine-Tuning**   Following work by Lee et al. (2023) and Panigrahi et al. (2023), we fine-tune the parameters within the localized components. For editing, we use a loss function $L = \lambda_1 L_{\text{injection}} + \lambda_2 L_{\text{retain}} + \lambda_3 L_{\text{SFT}}$, where $L_{\text{injection}}$ is a cross-entropy loss on the forget facts maximizing the probability of the alternative new false target. $L_{\text{retain}}$ is a cross-entropy loss on a train split of the remaining facts, and $L_{\text{SFT}}$ is a cross-entropy loss on the Pile dataset (Gao et al., 2020). We sweep over learning rates and injection loss $\lambda$s for three representative localizations in Appendix A.6 .

## 3 EDITING EVALUATION

In this section, we show the results of model editing with localization: we test localization techniques from Section 2.2, and edit these localized components using fine-tuning (Section 2.3). Here we focus on four main editing tasks: **sports-athlete-editing**, **full-sports-editing**, **counterfact-editing**, and **sequential-counterfact-editing**. We present augmenting results for **sports-unlearning** in Appendix A.1.

All the editing tasks are assessed based on prompt-completion based and adversarial relearning evaluations.

### 3.1 PROMPTING-BASED EVALUATION

Our prompting-based evaluation assesses an editing method's ability to forget or edit specific information while retaining unrelated knowledge. This is measured by evaluating how the model post-editing completes the prompts coming from the forget set: we report how accurately it recalls the undesired forgotten answer (*forget accuracy*), and how accurately it recalls the new desired edited answer (*edit accuracy*). In addition, we also measure the accuracy on facts not in the forget

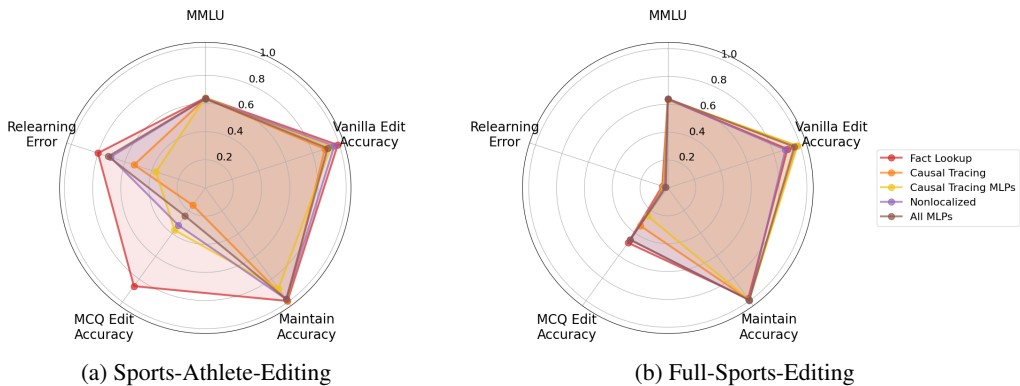

(a) Sports-Athlete-Editing  (b) Full-Sports-Editing

Figure 2: Spider plots illustrating the advantages of FLU for editing Sports across adversarial prompting and relearning evaluations. **(Left)** The Sports-Athlete-Editing plot shows that FLU localization leads to editing that is the most robust against MCQ prompting and relearning. **(Right)** The plot shows that most localizations perform approximately equivalently in the Full-Sports-Editing task, with FLU localization slightly better for MCQ.

set (*maintain accuracy*). In cases where a positive result is lower accuracy, we use the term *error* to denote 1 - accuracy (e.g. *forget error = 1 - forget accuracy*). Thus, well edited models should decrease forget accuracy, increase forget error, and increase edit accuracy. Results of these standard evaluations are reported in Appendix A.7.1.

Inspired by Patil et al. (2023) and Lynch et al. (2024), to ensure the editing process has not overfit to the specific format of the original prompts, we incorporate a robustness check using a multiple-choice question format (MCQ accuracy). This helps determine to what extent the model edited the information, and whether it can still access and utilize that knowledge when prompted differently. In this MCQ evaluation, the prompt also includes some in-context examples of answering multiple choice questions correctly. On the forget set, we refer to the accuracy of the model answering with the ground truth as the *MCQ Forget Accuracy* (stronger methods should decrease *MCQ Forget Accuracy*), and the accuracy of the model answering with the new edited answer's choice as the *MCQ Edit Accuracy* (stronger methods should increase *MCQ Edit Accuracy*).

Finally, we also evaluate the models' accuracy post-editing on MMLU (Hendrycks et al., 2021) as a proxy for general language understanding to measure any unintended side effects.

### 3.1.1 SPORTS TASKS

**Sports Prompting**  For the sports dataset, following Nanda et al. (2023), we first evaluate the accuracy of our models to complete the prompt, "Fact: [athlete] plays the sport of", with a one-shot example of Tiger Woods playing golf given first. Note that this is the same prompt used for the editing loss in the first place. For the MCQ evaluation, we use choices of all four sports (football, baseball, basketball, and golf).

We average accuracies over all models, for Sports-Athlete-Editing we average over editing both 16 and 64 facts, and for Full-Sports-Editing we average over editing Basketball, Baseball, and Football.

**Sports Results**  Our analysis reveals that editing employing FLU localization exhibits superior performance in forgetting the original information and adopting the edited information, across different prompt formats. As explained in Section 3.1, better editing should result in higher *MCQ Forget Error*, and higher *MCQ Edit Accuracy*. Figure 3 shows that FLU localized models are the best on both fronts. The difference is especially significant in Figure 3a, where only FLU model edits generalize meaningfully to MCQ, exceeding other localization methods by more than 40% in MCQ Edit Accuracy.

A comprehensive comparison of all localization methods with multiple-choice prompting is available in Appendix A.7.2, further supporting our findings.

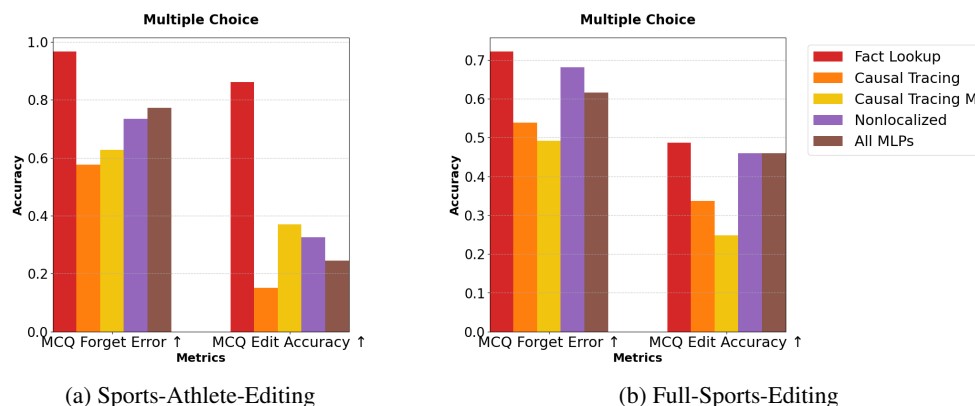

(a) Sports-Athlete-Editing          (b) Full-Sports-Editing

Figure 3: Bar charts showing results of MCQ evaluations, reporting both the forget error and edit accuracy when prompted with MCQ. For both **(a)** Sports-Athlete-Editing and **(b)** Full-Sports-Editing, FLU localization answers with the original answer the least (MCQ Forget Error) and answers with the edited answer most accurately (MCQ Edit Accuracy).

### 3.1.2 COUNTERFACT TASKS

**CounterFact Prompting** For CounterFact, we create an MCQ evaluation with four choices for every question, randomly ordering the true answer, the injected false answer, and two other question-specific LLM-generated incorrect answers. We also consider the original robustness and side effect evaluations from the Meng et al. (2023) dataset: the Paraphrase and Neighborhood facts accuracies. Answers of edited facts are meant to generalize to the Paraphrase evaluation, which phrases the fact in a different but equivalent way, so we report the *Paraphrase Edit Accuracy* (stronger methods should increase Paraphrase edit accuracy). Editing should not generalize to the Neighborhood evaluation, which presents similar but unrelated facts to the forget set, so we report the *Neighborhood Edit Error* which is lower if models incorrectly report the edited answer in these unrelated facts.

We again also use an MCQ evaluation, where the choices consist of the true answer, the injected false answer, and two other question-specific LLM-generated incorrect answers. We note that Paraphrase and Neighborhood evaluations ask for the answer in the same original format, so they are more in-distribution than MCQ. We average accuracies over all models, and for CounterFact-Editing we average over editing both 16 and 64 facts.

**CounterFact Results** Figure 5 illustrates the robustness of FLU editing in the MCQ and Paraphrase evaluations. Edited models using FLU localization answer less frequently with the original, incorrect information (*MCQ Forget Error*) and more frequently provide the new, edited answer. Furthermore, the *Neighborhood Edit Error* highlights that other localization methods exhibit slightly more pronounced side effects, inadvertently editing unintended, semantically similar facts.

Interestingly, sequential editing displays marginally greater robustness than nonsequential editing in MCQ when comparing between CounterFact-Editing and Sequential-CounterFact-Editing bars. This observation supports an approach that editing large sets of facts can be made more effective by partitioning the set and applying edits sequentially. We present results comparing all localizations across each prompt robustness evaluation in Appendix A.7.2, with the largely consistent conclusions.

### 3.2 ADVERSARIAL RELEARNING EVALUATION

We measure the ability of our models to withstand adversarial relearning, both to address the scenario in which adversaries may have fine-tuning access and as a measure for the quality of editing. We replicate the methodology of Deeb & Roger (2024), splitting our forget sets in two independent halves, retraining with half of the ground truth labels, and evaluating on the other half. This methodology aims to discern whether the editing technique successfully removed the underlying

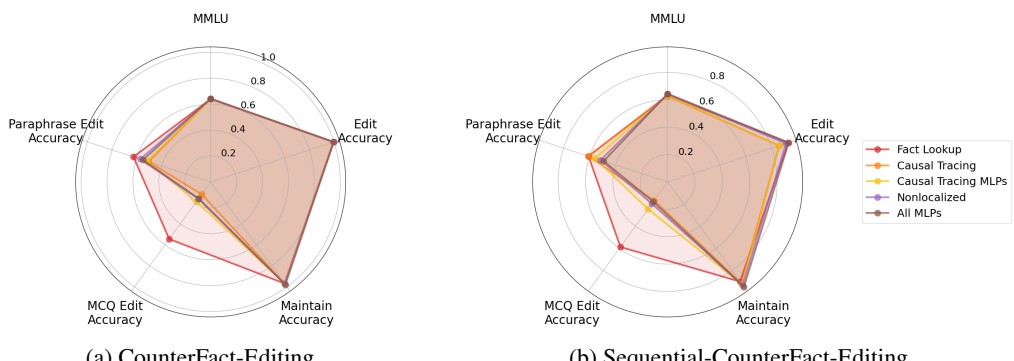

(a) CounterFact-Editing

(b) Sequential-CounterFact-Editing

Figure 4: Spider plots illustrating the advantages of FLU for editing CounterFact across prompting evaluations. **(Left)** The CounterFact-Editing plot shows that FLU localization leads to editing that is the most robust against MCQ prompting and Paraphrasing. **(Right)** The Sequential-CounterFact-Editing plot shows that FLU localization is the most robust against MCQ prompting.

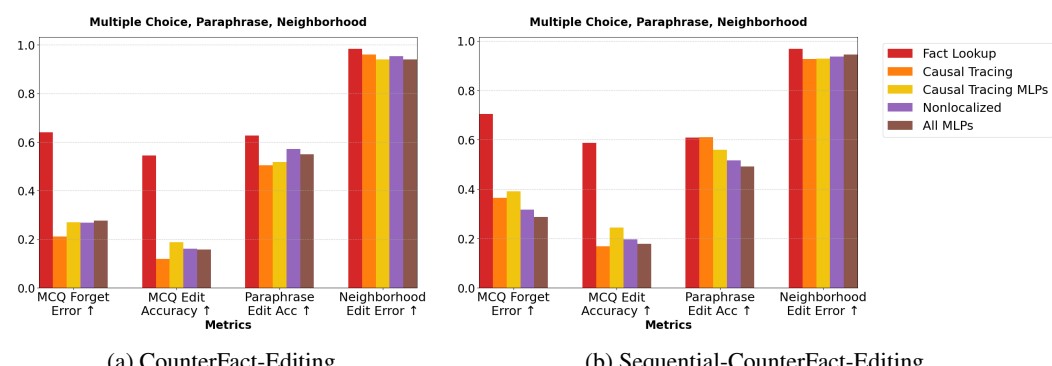

(a) CounterFact-Editing

(b) Sequential-CounterFact-Editing

Figure 5: Bar charts showing results of MCQ, Paraphrase, and Neighborhood prompt evaluations. For both **(a)** CounterFact-Editing and **(b)** Sequential-CounterFact-Editing, FLU localization has the most robust edit accuracy measured by MCQ and Paraphrase. FLU localization editing also does not incorrectly generalize to Neighborhood prompts. Sequential editing is slightly more robust than nonsequential editing in MCQ when comparing between CounterFact-Editing and Sequential-CounterFact-Editing bars.

factual association or merely obfuscated its direct retrieval while leaving it potentially susceptible to recovery when doing partial retraining.

We retrain with a rank-512 LoRA across all linear modules, with details available in Appendix A.7.3. In this section, we focus on the Sports-Athlete-Editing task, as in the other tasks it was either too easy to relearn (Full-Sports-Editing) or too hard to relearn any performance (all CounterFact tasks) across all localizations. Relearning isn't a valid evaluation for Full-Sports-Editing because the facts are not independent, and models should reasonably generalize from relearning on half the basketball athletes to answering with basketball on the other half of the basketball athletes. We show results from Counterfact and from all localizations in Appendix A.7.3.

We report averaged results over all models and editing both 16 and 64 athletes.

**Sports Results** Our adversarial relearning experiments, as depicted in Figure 6, reveal that retraining on a subset of the original "forgotten" data can recover a significant portion, as much as 63%, of the supposedly forgotten information when using OT methods like Causal Tracing and Causal Tracing MLPs. This suggests that these methods may simply mask direct retrieval of this information, leaving the model susceptible to this information recovery through retraining. In contrast, FLU localization exhibits greater resilience to such adversarial relearning, with only about 20% of

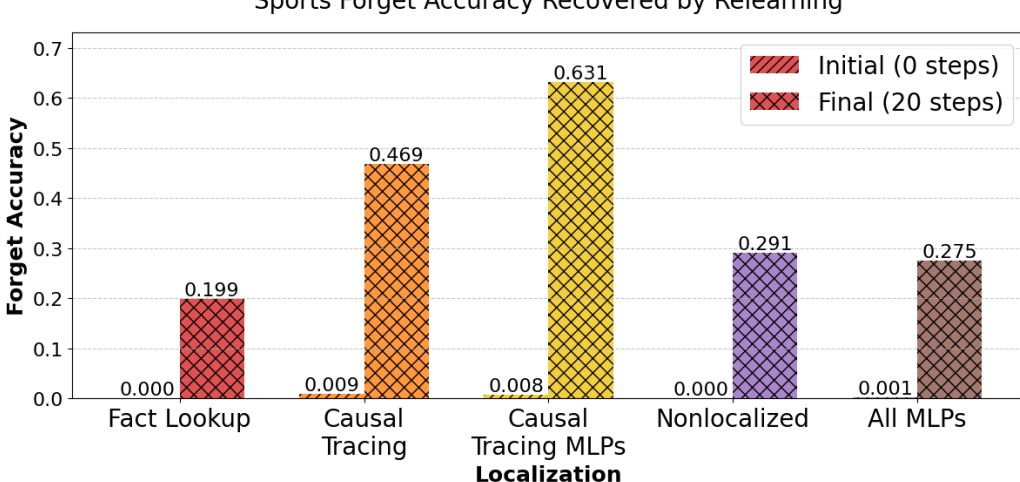

Figure 6: Relearning recovers the least accuracy on the forget set using FLU localizations. Relearning recovers significant accuracy on the original forget set in OT localizations (Causal Tracing and Causal Tracing MLPs).

the forgotten information recovered. This indicates that FLU localization may be more effective in targeting and removing the underlying knowledge, making it harder to recover through retraining.

### 3.3 LATENT KNOWLEDGE ANALYSIS

In this section, we provide more evidence of our hypothesis that FLU unlearning targets the source of intermediate latent knowledge. We analyze the Sports-Athlete-Editing task again here because the ground truth and the edited answers vary between one of only three possibilities.

We train logistic regression models (probes) (Alain & Bengio, 2018) on prompt activations following every model layer to predict the correct ground truth sport on the maintained set of athletes. This is possible because there are only three possible sports, so we can train binary classification probes for each sport and take the maximum classification over the sports. This is meant to discover internal representations of the true sport the model believes the answer to be: then, we apply these probes on the activations of the forget set of athletes.

We present graphs averaged over models and over 16 and 64 facts. Because layer is on the x-axis and models have different numbers of layers, averaging performance at a layer over multiple models isn't perfectly valid, so we also present probing graphs for each individual model and all localizations in Appendix A.7.4.

**Sports Results** In Figure 7, the probes on FLU consistently predict the forget answer less and the edit answer more than in any other localization, especially in early layers. Furthermore, the FLU probe classifications for the most part monotonically converge from their nonzero starting accuracy to 0 (for forget accuracy) and 1 (for edit accuracy).

Every other localization has much higher peak probe classification forget accuracy in the early layers, especially the OT localizations which have peak classification forget accuracy of almost 100%. This strongly suggests that these models still significantly represent the ground truth answer rather than the edit answer in early layers.

### 3.4 THE ROLE OF PARAMETER COUNT

In this section, we perform weight-masking to quantify the size of edits with different localizations, as well as to investigate which factual mechanisms are targeted when editing with different localizations. We employ a weight masking technique involving training a binary differentiable mask over individual weights of the model within the localized components, inspired by weight prun-

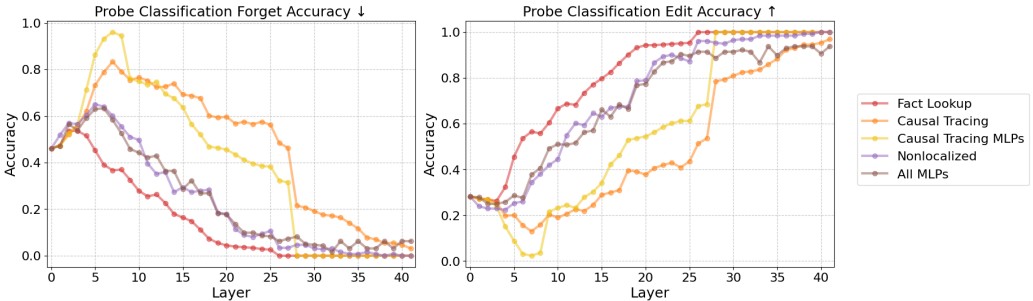

Figure 7: Linear probes applied to the forget set, classifying model activations after various layers. **(Left)** The line graph shows that some localizations still represent almost completely correct forget set knowledge in early layers, especially OT, while FLU localizations represent this original knowledge the least. **(Right)** The line graph shows that FLU localizations represent the edited rather than original answer earlier and more consistently throughout layers than any other localization.

ing/masking work (Bayazit et al., 2023; Panigrahi et al., 2023). In this case, no weight updates are being performed. Rather, the mask turns a subset of the weights to zero.

**Controlling for Parameter Count**   Although we already standardize the number of trainable parameters in most localizations, we additionally investigate if FLU editing is better than other localization techniques when controlling for the exact number of parameters that are masked.

We perform weight masking on the Sports-Unlearning, Sports-Athlete-Editing, and CounterFact-Editing tasks. Detailed results are reported in Appendix A.4. We find that when controlling for the size of the localization FLU is consistently more robust when subject to our suite of evaluations.

**Other Localizations Affect the Extraction Mechanism**   After training weight masks, we analyze the proportion of each mechanism (fact lookup, attribution extraction) that is masked by each localization's weight mask in Appendix A.5. We demonstrate that OT methods and nonlocalized editing all modify a higher proportion of the extraction head/MLP parameters than the fact lookup mechanism parameters, supporting our claim that OT methods target extraction mechanisms rather than the fact lookup mechanisms needed for robustness.

## 4  DISCUSSION

Recent work by Hase et al. (2023) argued that localization is not useful for model editing. Our findings demonstrate that the relationship between localization and fact editing/unlearning is more nuanced, and reveals that not all localization techniques are equal.

Our work evaluates the efficacy of different localization methods for modifying factual associations. We demonstrate clear benefits of localization for editing robustness through localized fine-tuning combined with FLU interpretability.

In Sections 3.3 and 3.4 and Appendix A.5, we provide evidence that OT and baseline approaches fail to be robust because they target easily-localizable and high direct logit importance extraction components that transform existing latent factual knowledge to the desired output format. This can fail to generalize to different input and output formats and does not target the source of knowledge in the model: other input/output formats can allow alternative mechanisms to extract this knowledge, and relearning can repair the original mechanisms to recover accuracy. In contrast, FLU mechanistic understanding allows us to target editing at the sites where knowledge is sourced, which robustly prevents that information from entering the latent stream in any format.

Our work also suggests unlearning/editing as a potential testbed for different interpretability methods, which might sidestep the inherent lack of ground truth in interpretability (Templeton et al., 2024). We hope our work provides a framework for evaluating localizations and explanations.

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

# A APPENDIX

## A.1 SPORTS UNLEARNING RESULTS

For unlearning on the Sports-Unlearning task, we use a loss function

$$L = \lambda_1 L_{\text{forget}} + \lambda_2 L_{\text{retain}} + \lambda_3 L_{\text{SFT}},$$

where $L_{\text{forget}}$ is an unlearning loss on the $D_{forget}$ subset of facts we want to forget, $L_{\text{retain}}$ is a cross-entropy loss on the remaining facts, and $L_{\text{SFT}}$ is a cross-entropy loss on the Pile dataset (Gao et al., 2020). The unlearning loss $L_{\text{forget}}$ we use is the $\log(1-p)$ measure (where p is the probability of the correct sport) from Mazeika et al. (2024) due to its empirical stability and fewer side effects: vanilla gradient ascent more strongly incentivizes the model to have significantly lower logprobs than wouldn't be encountered in a model that has not been trained on the factual association, and it detracts from model maintenance of $L_{\text{retain}}$ and $L_{\text{SFT}}$.

We present results on using various localizations on the Sports-Unlearning task in Table 1. The FLU localization allows unlearning to be more robust to the MCQ prompt format while maintaining performance on the MMLU dataset.

Table 1: Localized fine-tuning accuracy on standard evaluations: unlearning all basketball athletes and retaining all other facts.

| LOCALIZATION | FORGET ↓ | RETAIN ↑ | MCQ ↓ | MMLU ↑ |
|---|---|---|---|---|
| ATTRIB. PATCHING | **0.000** | **1.000** | 0.767 | 0.602 |
| CAUSAL TRACING | 0.201 | 0.998 | 0.849 | 0.611 |
| FLU | 0.002 | 0.995 | **0.110** | **0.613** |
| RANDOM | 0.952 | 0.980 | 0.822 | 0.612 |
| ALL-MLPS | **0.000** | 0.994 | 0.279 | 0.606 |
| NONLOCALIZED | **0.000** | 0.985 | 0.196 | 0.595 |

## A.2 FLU INTERPRETABILITY ANALYSIS

### A.2.1 SPORTS FACTS

We redo analysis from Nanda et al. (2023) on Gemma-7B, Gemma-2-9B, and Llama-3-8B. We train logistic regression models ("probes") to predict the correct sport given the internal representation of the model at a layer. We find that probes predicting the correct sport increase in accuracy significantly in layers 2 through 7 in Gemma-7B and 2 through 8 for Gemma-2-9B and Llama-3-8B (Figure 8).

Unlike Nanda et al. (2023), however, we find attention heads past layer 2 that impact the linear representation of attributes and thus could potentially be important for fact lookup. However, because they could likely play a variety of other different roles such as token concatenation, following the findings of Geva et al. (2023); Nanda et al. (2023) that MLPs do primary factual representation enrichment, in this work we only consider the MLPs as our localization.

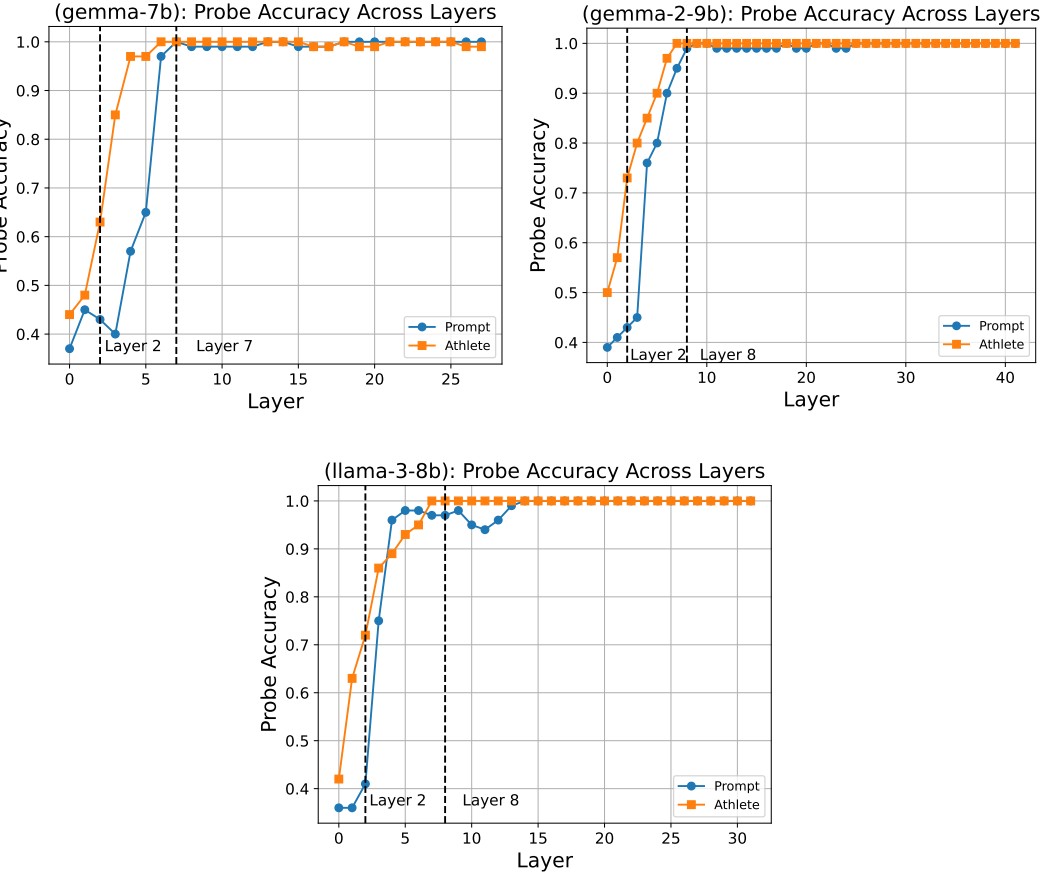

Figure 8: Accuracy of probes predicting the correct sport across layers for different models.

### A.2.2 COUNTERFACT

We repeat analysis from Nanda et al. (2023) and Geva et al. (2023) on Gemma-2-9B. We first measure the effect on the difference in logits between correct and incorrect answers of facts when patching the direct path of attention heads and MLPs to the final output, shown in Figure 9. An attention head or MLP will have a large effect on the logit difference if it is important in moving the factual information to the last token position or decoding it into the correct answer. We call these components part of the "fact extraction mechanism", and aim to find the source of the factual information moved by this mechanism.

To find this source, we patch the outputs of MLPs to this "fact extraction mechanism" and measure the resultant change in logit difference (Figure 10). An MLP would cause a large change in logit difference if it caused relevant representations to form that are then moved by the "fact extraction heads" to increase the probability of the correct output. We provide the logit differences for all 64 facts along with just the first 16 facts, and see that the logit differences are similar across the dataset splits. We take the MLPs with the highest change ($> 0.02$) and include them in our FLU localization of CounterFact.

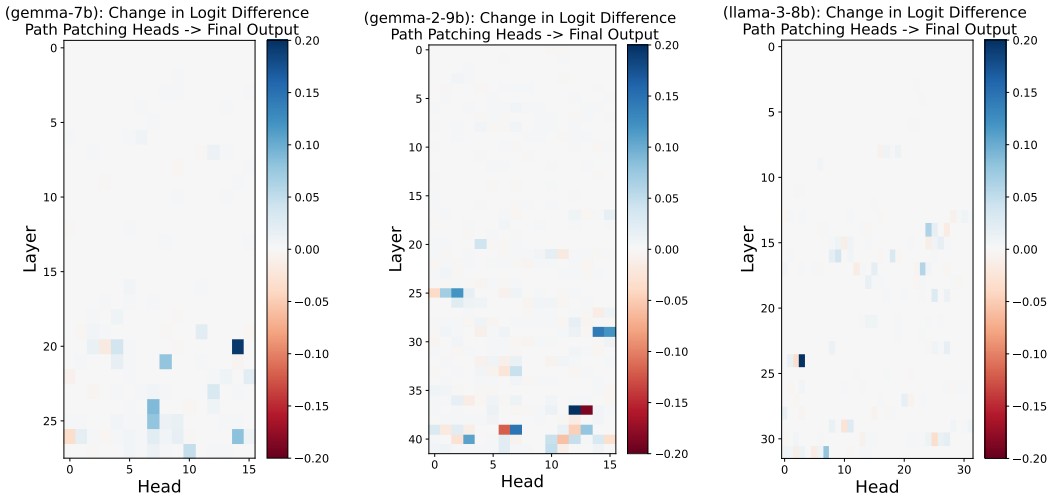

Figure 9: Change in logit difference when patching attention heads to the output.

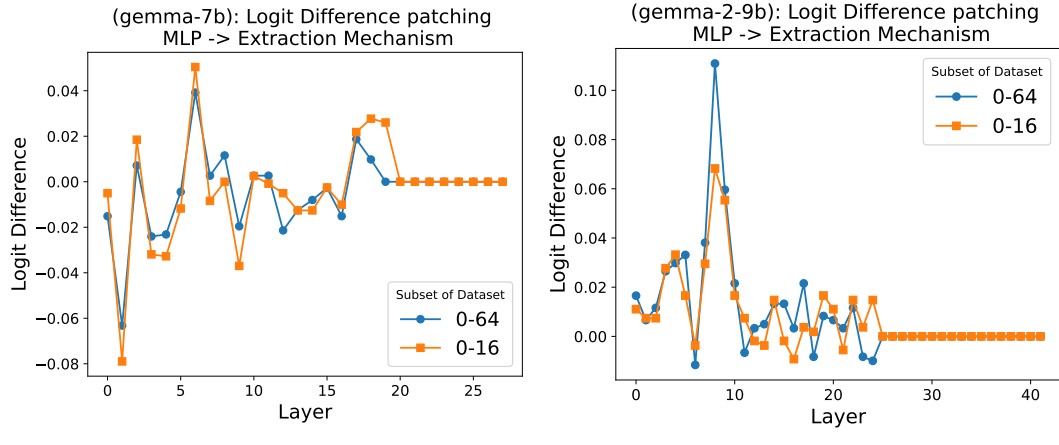

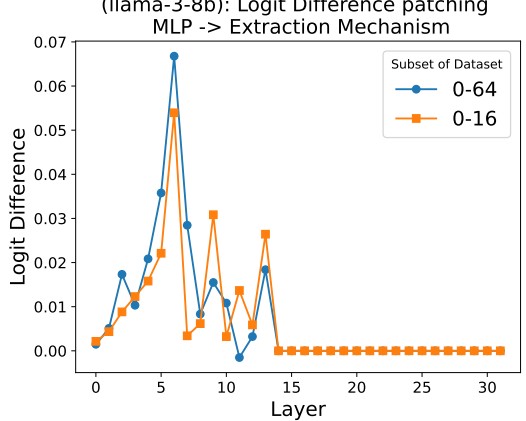

Figure 10: Logit difference when patching MLPs to the extraction mechanism found above for different models.

### A.3 OT Selected Components

What MLPs do the automated OT methods localize? We explore the attribution scores of the automated localization methods (causal tracing and attribution patching) on the MLPs to see if automated localization methods can detect the FLU mechanism. In Figures 11 and 14, for Gemma-7B, we see that both CT and AP localizations target the later layer MLPs instead of the FLU mechanism (Figure 11).

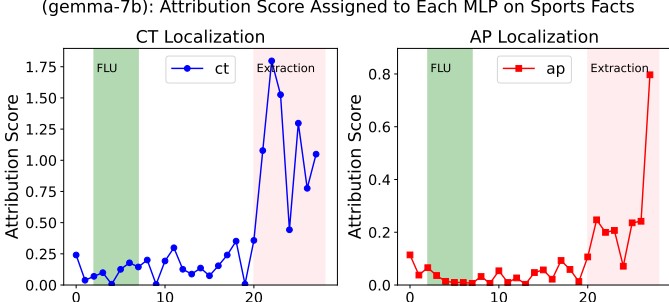

Figure 11: Attribution scores on MLPs on sports facts for Gemma-7B.

For AP localization, this trend continues with Gemma-2-9B (Figure 12) and Llama-3-8B (Figure 13). However, CT localization does highlight some of the early layer MLPs that are in the FLU mechanism, especially for Gemma-2-9B.

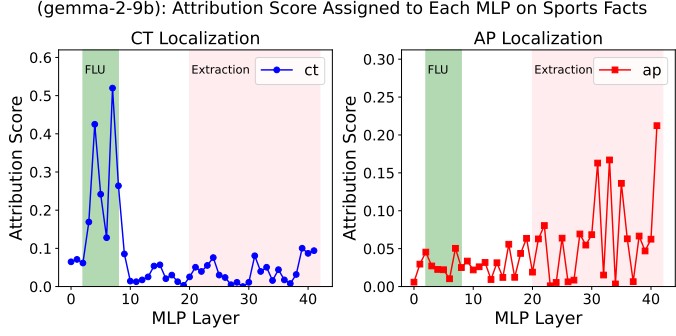

Figure 12: Attribution scores on MLPs on sports facts for Gemma-2-9B.

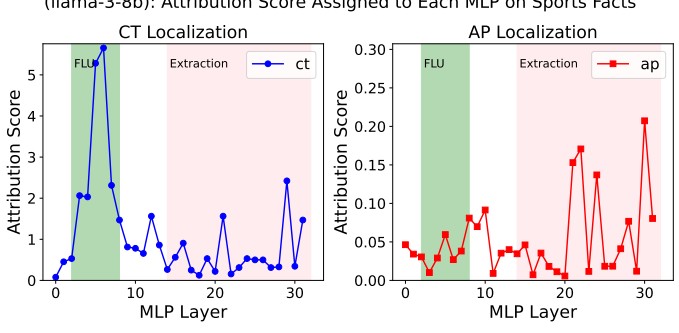

Figure 13: Attribution scores on MLPs on sports facts for Llama-3-8B.

We repeat this analysis on CounterFact in Figures 14 to 16. Again we see AP localizations in particular assign higher scores to later-layer MLPs, and CT only highlights FLU components on Gemma-2-9B, localizing other extraction layers on the other models.

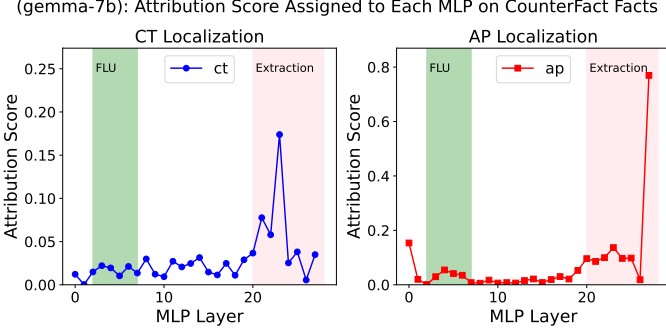

Figure 14: Attribution scores on MLPs on CounterFact facts for Gemma-7B.

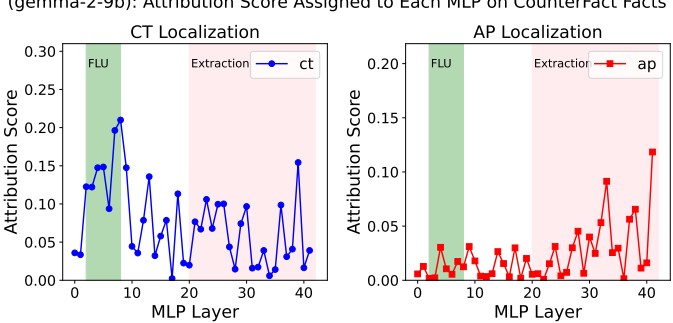

Figure 15: Attribution scores on MLPs on CounterFact facts for Gemma-2-9B.

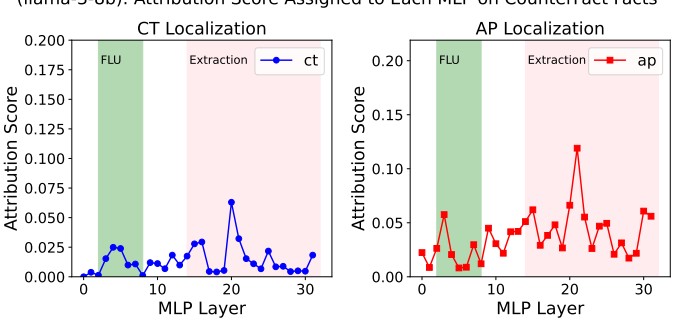

Figure 16: Attribution scores on MLPs on CounterFact facts for Llama-3-8B.

## A.4 WEIGHT MASKING

In this section we employ weight masking to quantify the size of weight updates needed to un-learn/edit facts, for more direct comparisons. Our loss function $L = \lambda_1 L_{\text{forget}} + \lambda_2 L_{\text{retain}} + \lambda_3 L_{\text{SFT}} + \lambda_4 L_{\text{reg}}$ now includes an L1 regularization term to control the sparsity. We empirically evaluate how a learned binary mask over individual weights of the localized components can produce editing/unlearning, and vary the size of this mask. "Manual Interp" refers to the FLU localization technique for all the following results in this section.

### A.4.1 UNLEARNING SPORTS

We show standard evaluations across a sweep of discretization thresholds, which directly corresponds to the size of the model edit. Figure 17 shows the accuracy on the forget and retain sets for unlearning basketball across different edit sizes. Here, we see all methods being effective in

unlearning basketball facts while retaining all other facts. While AP and CT localizations cause the model to have zero accuracy on the in-distribution set with much fewer masked weights needed, when checking for generalization using a multiple-choice format we clearly see that only manual localization has successfully generalized the unlearning of basketball facts (Figure 17, right).

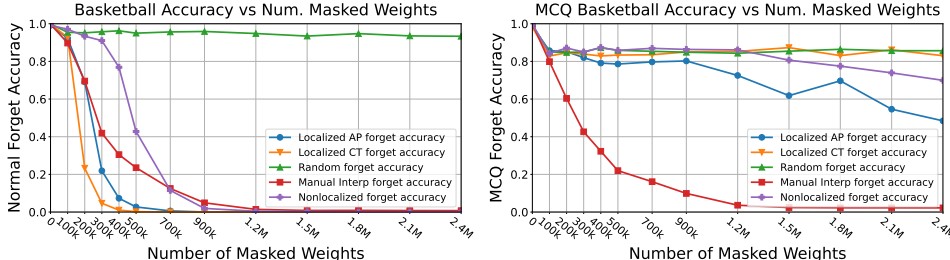

Figure 17: **(Left)** Testing the models' unlearning of basketball athletes against the number of weights masked. **(Right)** Testing the models' unlearning of basketball athletes against the number of weights masked, in the MCQ prompt format.

We find similar results when testing for performance degradation on MMLU (because we have to evaluate many model variations, we use a smaller MMLU test set from Polo et al. (2024)). While all localized methods perform well when evaluated normally (Figure 18, left), Figure 18 (right) shows manual localization generalizes for minimizing loss of MMLU capabilities while unlearning sports facts in the MCQ format compared to the other methods.

### A.4.2 EDITING ATHLETES

For editing the subset of athletes, Figure 19 shows that causal tracing localization causes the model to have 0% accuracy on the forget set, and FLU and nonlocalized editing cause the model to have near guessing rate (33%) accuracy. However, only manual localization minimizes loss of capabilities while editing the athlete subset (Figure 20).

Furthermore, no other method completely generalizes this unlearning to the MCQ prompt format (Figure 19), and manual localization remains superior in minimizing loss of capabilities while unlearning the athlete subset (Figure 20, right).

### A.4.3 EDITING COUNTERFACT

We find similar results for editing on the CounterFact dataset. However, we find minimal difference in MMLU accuracy in all methods at all numbers of masked weights. Thus, we instead report the maintain and forget accuracies of these facts at different discretization thresholds in Figure 21.

Additionally, we report a comparison of all localizations across discretization thresholds for normal and MCQ forget sets in Figure 22 and Figure 23. We see that FLU outperforms all other methods of localization in preserving maintain accuracy while decreasing forget accuracy.

We perform additional adversarial analysis of accuracies across different discretization thresholds. We report the "paraphrase" and "neighborhood" adversarial results in Figure 24 and Figure 25, but find no significant results.

### A.5 MECHANISM WEIGHT ANALYSIS

We analyze the actual components localized by each localization type and our baselines, for the CounterFact editing task. We seek to demonstrate that the OT localizations and baselines target extraction mechanisms rather than just the FLU mechanisms.

First, in Table 2, we compare the parameter counts of the part of each mechanism that is present in each localization. Table 2 shows that causal tracing and attribution patching both have the potential to modify a considerable proportion of the extraction heads and extraction MLPs.

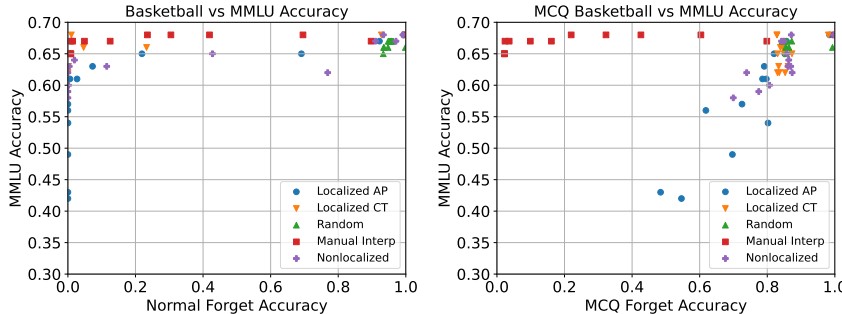

Figure 18: Unlearning basketball facts. **(Left)** Measuring MMLU and forget set performance across different discretization thresholds. **(Right)** Measuring MMLU and MCQ forget set performance across different discretization thresholds.

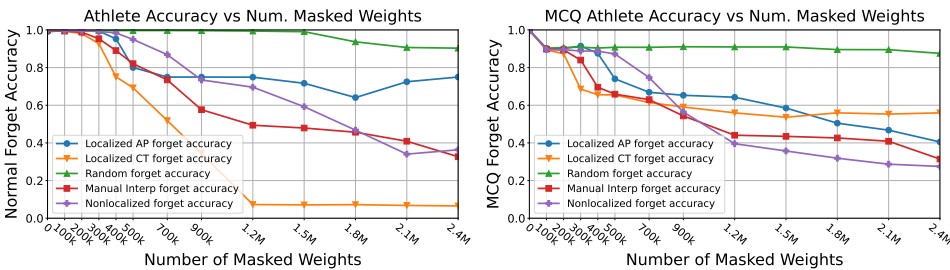

Figure 19: Editing subset of athletes. **(Left)** Measuring accuracy on the forget set. **(Right)** Measuring accuracy on the forget set in the MCQ prompt format.

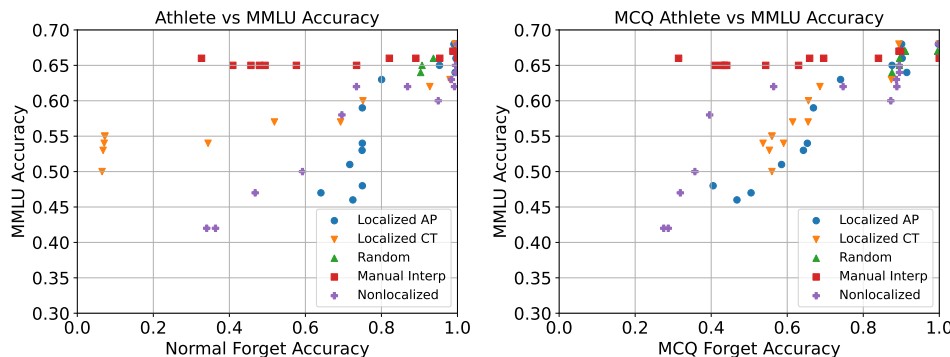

Figure 20: Editing subset of athletes. **(Left)** Measuring MMLU and forget set performance across different discretization thresholds. **(Right)** Measuring MMLU and MCQ forget set performance across different discretization thresholds.

Then, in Table 3, we compare the proportion of each mechanism that is masked when using a localized weight mask and discretizing to about 6 million weights. This is one approximate metric for how much each mechanism is modified by the localized editing. Table 3 demonstrates that attribution patching, causal tracing, and nonlocalized editing all modify a higher proportion of the extraction head/MLP weights than the fact lookup mechanism weights.

This supports our argument that OT methods target high logit-diff extraction mechanisms, rather than the fact lookup mechanisms that enrich the latent stream with the correct attributes, which decreases the robustness of edits/unlearning. It is important to note that since our FLU localization is based on our discovered mechanisms, this does not serve as an evaluation of FLU (since by definition FLU localization will only localize the FLU mechanism), but rather only of causal tracing and attribution patching.

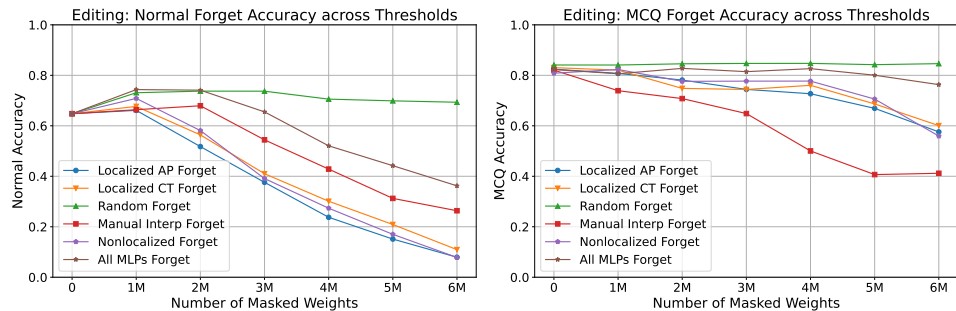

Figure 21: Editing CounterFact facts. **(Left)** Testing models accuracy on the normal forget set. **(Right)** Testing the models' accuracy in the MCQ prompt format.

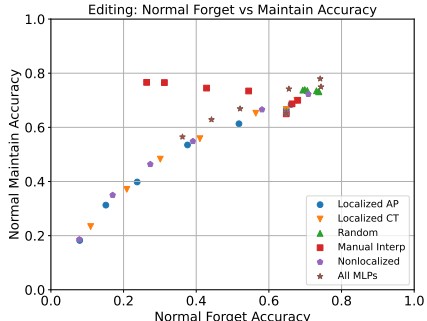

Figure 22: Accuracy on normal forget set vs on the maintain set across localizations and discretization thresholds.

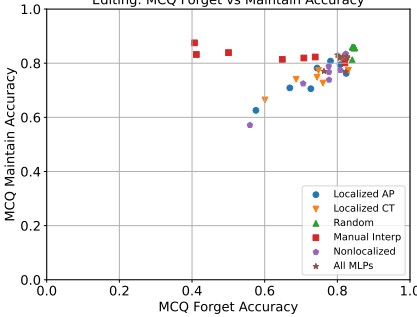

Figure 23: Accuracy on multiple choice input vs on the maintain set across localizations and discretization thresholds.

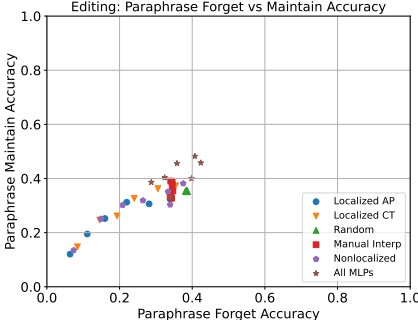

Figure 24: Accuracy on paraphrased input vs on the maintain set across localizations and discretization thresholds.

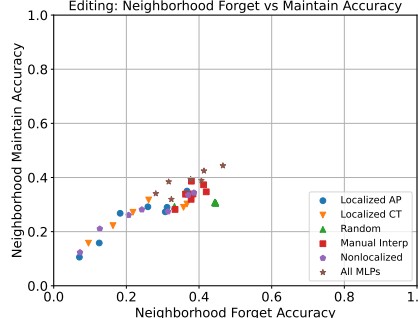

Figure 25: Accuracy on "neighborhood" input vs on the maintain set across localizations and discretization thresholds.

## A.6 HYPERPARAMETERS

Across all tasks except Sequential-CounterFact-Editing and all models, we fine tune using 50 iterations of batch size 4 with 16 accumulation steps, using an AdamW optimizer (Kingma & Ba, 2017) with 0 weight decay and a cosine annealing scheduler. For Gemma-2-9b, we are forced to use an 8-bit optimizer to fit our training in the memory of 1 GPU. We find that the optimal learning rate is quite sensitive to the localization used and the edit task, so we first sweep over learning rates to find reasonable learning rates. We sweep over the learning rates of 2e-6, 5e-6, 1e-5, 2e-5, 5e-5, and 1e-4, training models over 50 iterations with all $\lambda$s set to 1. We also tune the $\lambda_1$ parameter associated with the $L_{\text{injection}}$ cross entropy loss. We don't tune the other $\lambda$ parameters because they are all

Table 2: Comparison of total parameters of each mechanism that are present in each localization, for editing 16 facts from CounterFact

| LOCALIZATION | EXTRACTION HEADS | EXTRACTION MLPS | FACT LOOKUP |
|---|---|---|---|
| TOTAL | 27,448,320 | 1,027,604,480 | 1,130,364,928 |
| ATTRIB. PATCHING | 13,724,160 (50.0%) | 616,562,688 (60.0%) | 102,760,448 (9.1%) |
| CAUSAL TRACING | 8,234,496 (30.0%) | 308,281,344 (30.0%) | 411,041,792 (36.4%) |
| FLU | 0 | 0 | 1,130,364,928 (100.0%) |
| ALL-MLPS | 0 | 1,027,604,480 (100.0%) | 1,130,364,928 (100.0%) |
| NONLOCALIZED | 27,448,320 (100.0%) | 1,027,604,480 (100.0%) | 1,130,364,928 (100.0%) |

Table 3: Comparison of parameters of each mechanism that are masked by a trained weight mask, discretized to about 6 million weights

| LOCALIZATION TYPE | EXTRACTION HEADS | EXTRACTION MLPS | FACT LOOKUP |
|---|---|---|---|
| **TOTAL (BASELINE)** | 27,448,320 (100%) | 1,027,604,480 (100%) | 1,130,364,928 (100%) |
| ATTRIB. PATCHING | 165,300 (0.60%) | 1,479,877 (0.14%) | 1,385,198 (0.12%) |
| CAUSAL TRACING | 30,828 (0.11%) | 1,491,040 (0.15%) | 1,424,059 (0.13%) |
| FLU | 0 (0.0%) | 0 (0.0%) | 6,248,039 (0.55%) |
| ALL-MLPS | 0 (0.0%) | 1,378,744 (0.13%) | 1,663,772 (0.15%) |
| NONLOCALIZED | 358,918 (1.3%) | 1,198,211 (0.12%) | 1,174,939 (0.10%) |

maintenance losses, and setting them to 1 works sufficiently to maintain performance across almost all setups.

For the Sequential-CounterFact-Editing task, we use the same hyperparameters from the CounterFact-Editing task and we train for 100 total iterations rather than 50, using 25 for each subset of 16 facts. We choose 25 iterations because it balances between being half the number of iterations we typically use per subset of that size, and also double the number of steps overall as we use in CounterFact-Editing.

To avoid leaking evaluation information through this sweep process, we optimize learning rate for the objective of $(1 - $ *Forget Set Ground Truth Accuracy*$) + $ *Forget Set Edit Accuracy* $+$ *Maintain Set Ground Truth Accuracy* $+$ *Pile Accuracy*, avoiding any of our robustness metrics (one could view this sweep process as simply another part of the training process, since we only use train-time information). We run sweeps for Causal Tracing, Manual Fact Lookup, and No Localization. We then use the hyperparameters from Causal Tracing for Attribution Patching and Random (which all localize to MLPs and attention components), we use the hyperparameters from Fact Lookup for Random MLPs, Causal Tracing MLPs, and Attribution Patching MLPs (all localize to the same number of MLPs), and we use the hyperparameters from No Localization for All MLPs (which have the largest number of active parameters).

### A.6.1 SAMPLE HYPERPARAMETER SWEEP

In Table 4, we show the full results of one sweep, optimizing learning rate for editing 16 facts from CounterFact. Especially for No Localization, some learning rates fail to edit in the correct answer with high accuracy, or fail to maintain accuracy on the maintain set. In Table 5, we see that editing results are not particularly sensitive to the coefficient used with the injection cross entropy loss.

### A.6.2 ALL HYPERPARAMETERS USED

Table 6 has all learning rates used and Table 7 has all injection loss coefficients used.

### A.7 EVALUATION DETAILS

### A.7.1 DETAILS ON STANDARD PROMPT EVALUATIONS

We report standard metrics of Forget Error, Edit Accuracy, and Maintain Accuracy, in the same prompt format that the models were trained on. These metrics are optimized by the loss, so we

Table 4: Gemma-7b learning rate sweep, editing 16 CounterFact facts.

| | Pile Accuracy ↑ | Forget Accuracy ↓ | Edit Accuracy ↑ | Maintained Accuracy ↑ | Overall Score ↑ |
|---|---|---|---|---|---|
| *FLU* | | | | | |
| LR 0.0001 | 0.488 | 0.000 | 1.000 | 0.698 | 3.186 |
| LR 1e-05 | 0.513 | 0.000 | 1.000 | 0.975 | 3.488 |
| LR 2e-05 | 0.542 | 0.000 | 1.000 | 0.950 | 3.492 |
| LR 2e-06 | 0.499 | 0.007 | 0.961 | 0.869 | 3.323 |
| LR 5e-05 | 0.520 | 0.000 | 1.000 | 0.822 | 3.342 |
| LR 5e-06 | 0.528 | 0.000 | 0.999 | 0.980 | **3.507** |
| *Localized CT* | | | | | |
| LR 0.0001 | 0.462 | 0.001 | 0.999 | 0.697 | 3.157 |
| LR 1e-05 | 0.513 | 0.000 | 1.000 | 0.984 | 3.496 |
| LR 2e-05 | 0.507 | 0.000 | 1.000 | 0.961 | 3.467 |
| LR 2e-06 | 0.540 | 0.036 | 0.921 | 0.849 | 3.274 |
| LR 5e-05 | 0.488 | 0.000 | 1.000 | 0.846 | 3.334 |
| LR 5e-06 | 0.537 | 0.000 | 1.000 | 0.982 | **3.519** |
| *Nonlocalized* | | | | | |
| LR 0.0001 | 0.062 | 0.032 | 0.557 | 0.094 | 1.680 |
| LR 1e-05 | 0.520 | 0.000 | 1.000 | 0.892 | 3.412 |
| LR 2e-05 | 0.479 | 0.001 | 0.998 | 0.807 | 3.284 |
| LR 2e-06 | 0.529 | 0.000 | 1.000 | 0.982 | 3.510 |
| LR 5e-05 | 0.046 | 0.034 | 0.710 | 0.092 | 1.815 |
| LR 5e-06 | 0.536 | 0.000 | 1.000 | 0.988 | **3.524** |

Table 5: Gemma-7b inject loss coefficient sweep, editing 16 CounterFact facts.

| | Pile Accuracy ↑ | Forget Accuracy ↓ | Edit Accuracy ↑ | Maintained Accuracy ↑ | Overall Score ↑ |
|---|---|---|---|---|---|
| *FLU* | | | | | |
| FC 0.1 | 0.538 | 0.000 | 0.998 | 0.990 | **3.525** |
| FC 0.2 | 0.510 | 0.000 | 0.999 | 0.981 | 3.491 |
| FC 0.5 | 0.508 | 0.000 | 0.999 | 0.985 | 3.493 |
| FC 1 | 0.510 | 0.000 | 0.999 | 0.973 | 3.483 |
| FC 2 | 0.532 | 0.000 | 1.000 | 0.984 | 3.515 |
| FC 5 | 0.534 | 0.000 | 1.000 | 0.985 | 3.518 |
| *Localized CT* | | | | | |
| FC 0.1 | 0.532 | 0.004 | 0.995 | 0.983 | 3.506 |
| FC 0.2 | 0.535 | 0.001 | 0.998 | 0.986 | **3.518** |
| FC 0.5 | 0.524 | 0.000 | 0.999 | 0.974 | 3.497 |
| FC 1 | 0.519 | 0.000 | 1.000 | 0.988 | 3.507 |
| FC 2 | 0.536 | 0.000 | 1.000 | 0.979 | 3.514 |
| FC 5 | 0.518 | 0.000 | 1.000 | 0.975 | 3.493 |
| *Nonlocalized* | | | | | |
| FC 0.1 | 0.525 | 0.001 | 0.998 | 0.969 | 3.492 |
| FC 0.2 | 0.562 | 0.000 | 0.999 | 0.982 | **3.543** |
| FC 0.5 | 0.524 | 0.000 | 1.000 | 0.962 | 3.486 |
| FC 1 | 0.531 | 0.000 | 1.000 | 0.980 | 3.511 |
| FC 2 | 0.514 | 0.000 | 1.000 | 0.978 | 3.492 |
| FC 5 | 0.528 | 0.000 | 1.000 | 0.975 | 3.503 |

expect all localizations to do almost perfectly on these evaluations. Figures 26 to 29 show that localizations perform approximately equivalently on these basic evaluations across all tasks.

### A.7.2   DETAILS ON ADVERSARIAL PROMPT EVALUATIONS

We report the adversarial prompt evaluations from Section 3.1 across all localizations. Figures 30 to 33 all show that FLU localization is more robust in MCQ compared to every other localization (significantly stronger for CounterFact). Figures 32 and 33 show that FLU localization is optimal in Paraphrase and Neighborhood in all cases except for Paraphrase compared to the Random localization in Sequential-CounterFact-Editing.

Table 6: Optimal learning rates for different models, task types, and localizations.

| Model | 64 athletes to random sport | Basketball Athletes to Golf | 16 CounterFact facts | 64 CounterFact facts |
|---|---|---|---|---|
| **Gemma** | | | | |
| FLU | $1 \times 10^{-5}$ | $2 \times 10^{-5}$ | $5 \times 10^{-6}$ | $1 \times 10^{-5}$ |
| Localized CT | $1 \times 10^{-5}$ | $5 \times 10^{-6}$ | $5 \times 10^{-6}$ | $2 \times 10^{-5}$ |
| Nonlocalized | $2 \times 10^{-6}$ | $5 \times 10^{-6}$ | $5 \times 10^{-6}$ | $2 \times 10^{-6}$ |
| **Gemma 2** | | | | |
| FLU | $1 \times 10^{-5}$ | $5 \times 10^{-5}$ | $5 \times 10^{-5}$ | $2 \times 10^{-5}$ |
| Localized CT | $5 \times 10^{-5}$ | $5 \times 10^{-5}$ | $5 \times 10^{-5}$ | $2 \times 10^{-5}$ |
| Nonlocalized | $2 \times 10^{-5}$ | $1 \times 10^{-5}$ | $5 \times 10^{-6}$ | $5 \times 10^{-6}$ |
| **Llama 3** | | | | |
| FLU | $5 \times 10^{-5}$ | $5 \times 10^{-5}$ | $1 \times 10^{-4}$ | $5 \times 10^{-5}$ |
| Localized CT | $1 \times 10^{-4}$ | $5 \times 10^{-5}$ | $2 \times 10^{-5}$ | $5 \times 10^{-5}$ |
| Nonlocalized | $2 \times 10^{-5}$ | $1 \times 10^{-5}$ | $2 \times 10^{-5}$ | $1 \times 10^{-5}$ |

Table 7: Optimal inject loss coefficients for different models, task types, and localizations.

| Model | 64 athletes to random sport | Basketball Athletes to Golf | 16 CounterFact facts | 64 CounterFact facts |
|---|---|---|---|---|
| **Gemma** | | | | |
| FLU | 5.0 | 5.0 | 0.1 | 2.0 |
| Localized CT | 0.1 | 1.0 | 0.2 | 1.0 |
| Nonlocalized | 0.2 | 1.0 | 0.2 | 1.0 |
| **Gemma 2** | | | | |
| FLU | 1.0 | 5.0 | 2.0 | 0.5 |
| Localized CT | 5.0 | 5.0 | 2.0 | 2.0 |
| Nonlocalized | 5.0 | 5.0 | 1.0 | 1.0 |
| **Llama 3** | | | | |
| FLU | 5.0 | 0.2 | 1.0 | 0.2 |
| Localized CT | 2.0 | 1.0 | 0.1 | 2.0 |
| Nonlocalized | 2.0 | 0.1 | 2.0 | 0.2 |

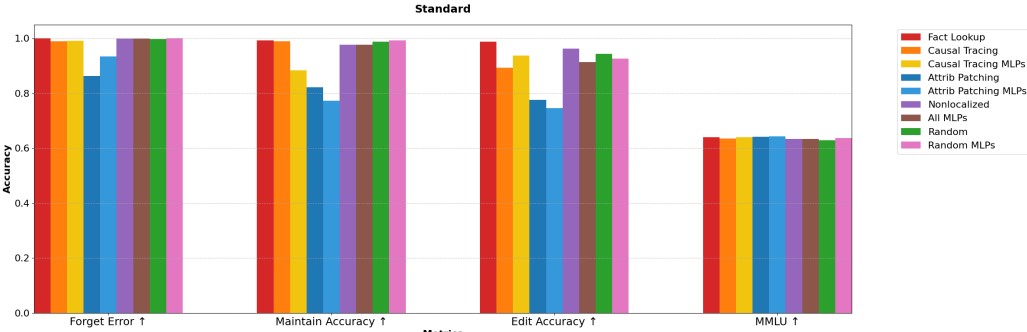

Figure 26: Standard Prompting results for Sports-Athlete-Editing, across all localizations.

### A.7.3 DETAILS ON ADVERSARIAL RELEARNING

We retrain the model for 20 iterations with cross-entropy on half of the forget set (along with a standard retain and SFT loss), adding up all losses with loss coefficient 1.

We present relearning results for all localizations averaged over models. As shown in Figure 34, the FLU localization remains optimal, although the baselines of Nonlocalized, All MLPs, Random, and Random MLPs are competitive.

We also present relearning results on the other tasks. As mentioned in Section 3.2, since Full-Sports-Editing forget facts are not independent, we don't expect valid results from relearning. Thus, in Figure 35, we see that every localization regains 100% editing accuracy.

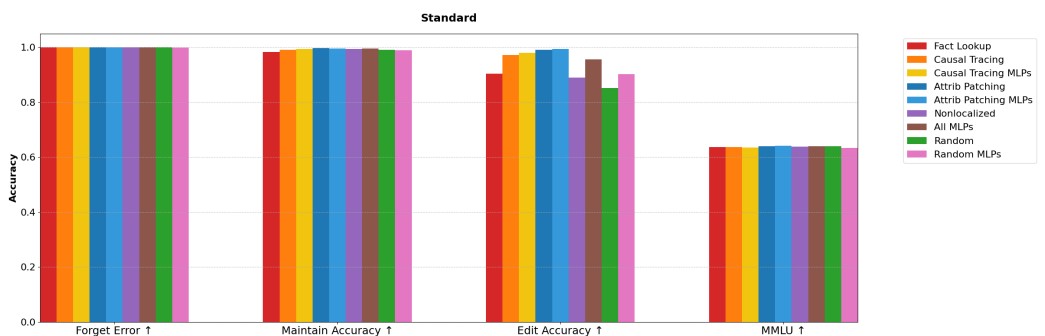

Figure 27: Standard Prompting results for Full-Sports-Editing, across all localizations.

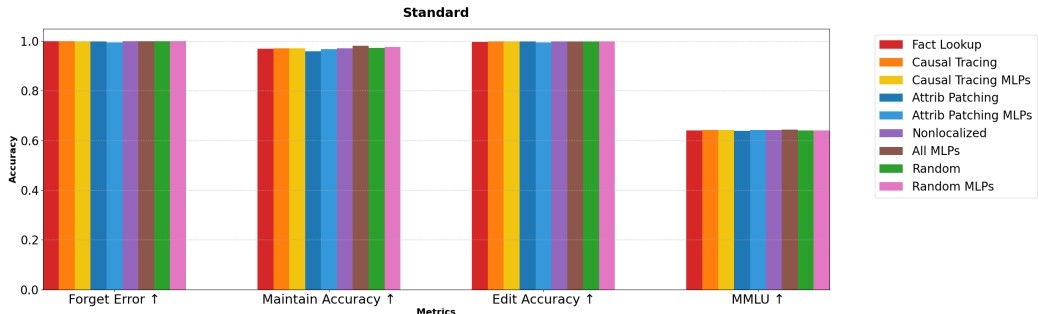

Figure 28: Standard Prompting results for CounterFact-Editing, across all localizations.

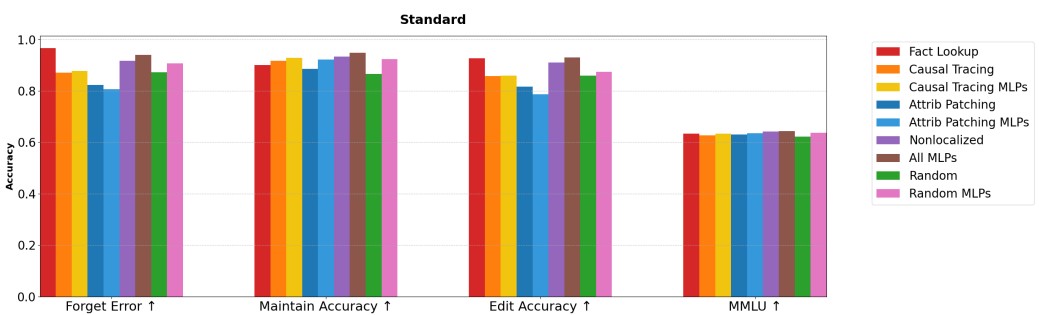

Figure 29: Standard Prompting results for Sequential-CounterFact-Editing, across all localizations.

On CounterFact-Editing and Sequential-CounterFact-Editing, as shown in Figures 36 and 37, none of the localizations relearn more than 7% accuracy, suggesting adversarial relearning was not a sufficiently strong enough evaluation for these tasks. Regardless, FLU localization is either the most or second-most robust localization to relearning, although localizations don't differ by much.

### A.7.4 DETAILS ON LATENT KNOWLEDGE

We present the probing classification accuracies for the three models separately here, as well as for all localizations we previously left out.

In Gemma-7b and Llama-3-8b, FLU probing is the most monotonic and the best in the early layers, either steadily decreasing to 0 for forget accuracy or increasing to 1 for edit accuracy, with the least extreme peaks. In Gemma-2-9b, the Nonlocalized, All MLPs, and Random MLPs baselines are competitive with FLU. The other OT localization, Attribution Patching, has 100% probing forget accuracy across many layers, suggesting it represents the ground truth answer very clearly.

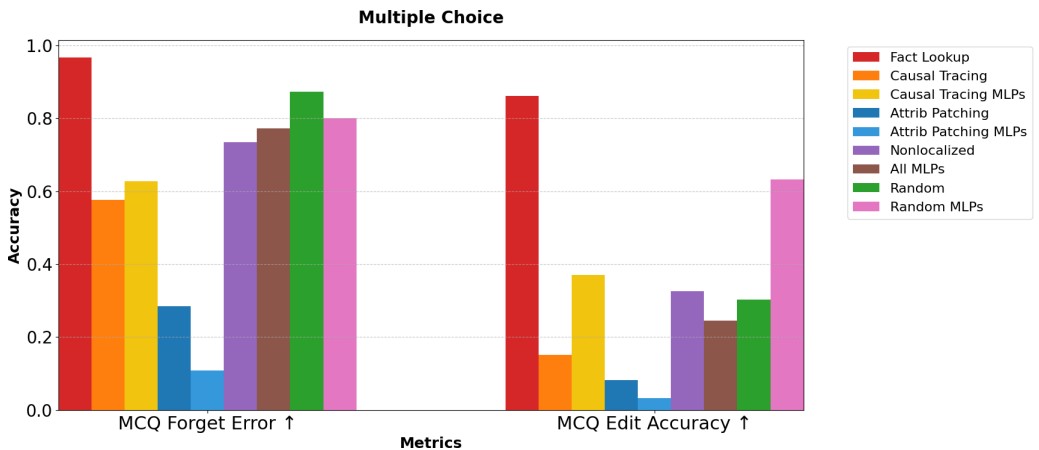

Figure 30: Adversarial Prompting results for Sports-Athlete-Editing, across all localizations.

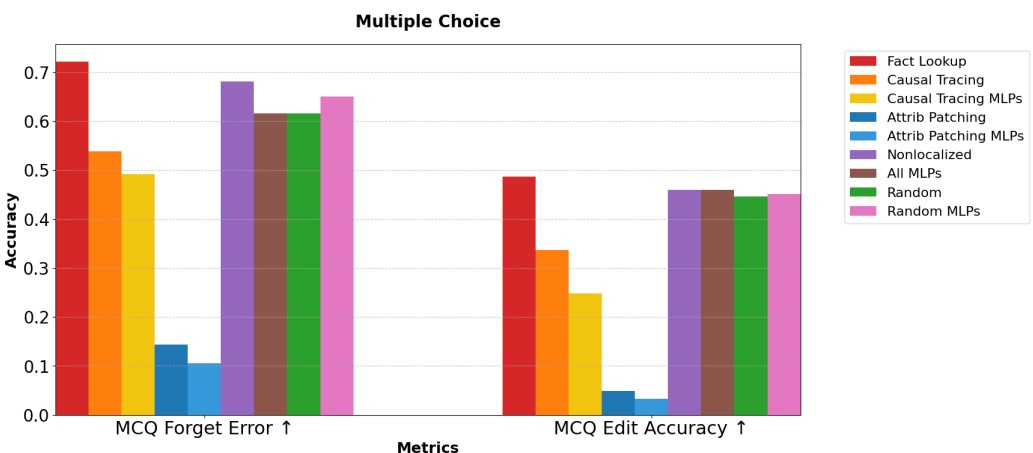

Figure 31: Adversarial Prompting results for Full-Sports-Editing, across all localizations.

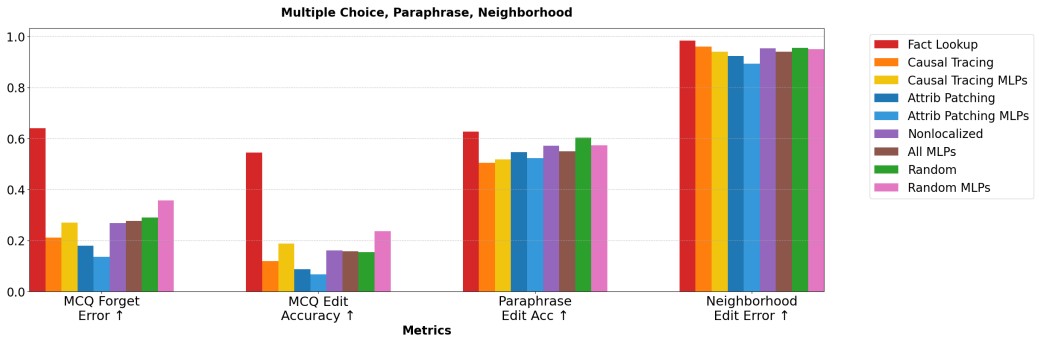

Figure 32: Adversarial Prompting results for CounterFact-Editing, across all localizations.

### A.8 SOFT PROMPT EVALUATIONS

Because many localizations seem to be weak to prompting schemes, we attempt a simple adaptive attack of soft prompts, where we optimize the continuous embeddings at the end of the prompt to recover the correct answer on half of our forget set. We then evaluate the model's performance on the other half, with this soft prompt in place (Lester et al., 2021). We average over evaluations from

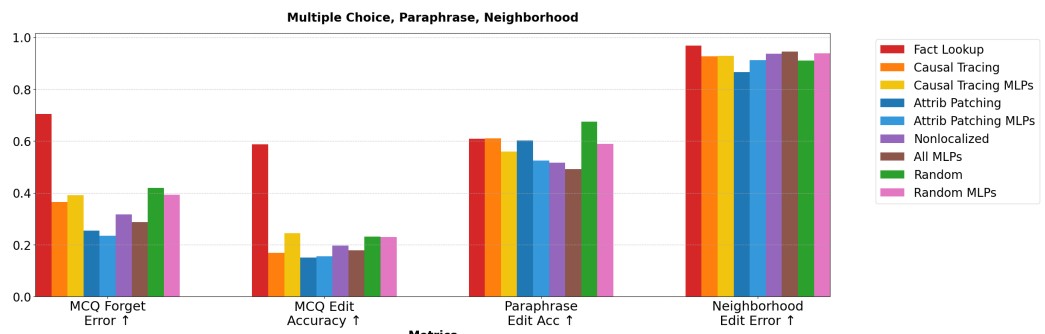

Figure 33: Adversarial Prompting results for Sequential-CounterFact-Editing, across all localizations.

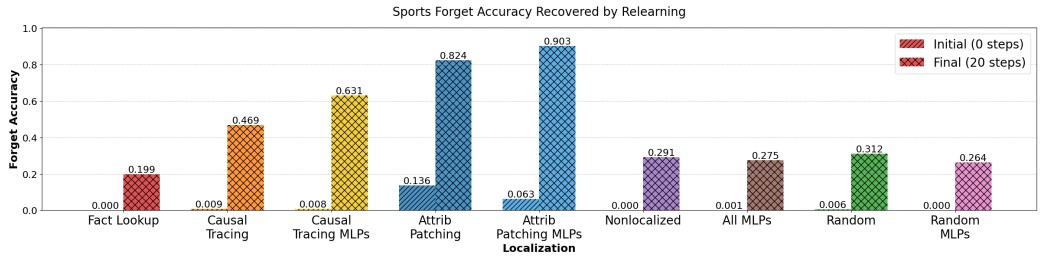

Figure 34: Relearning results for Sports-Athlete-Editing, across all localizations.

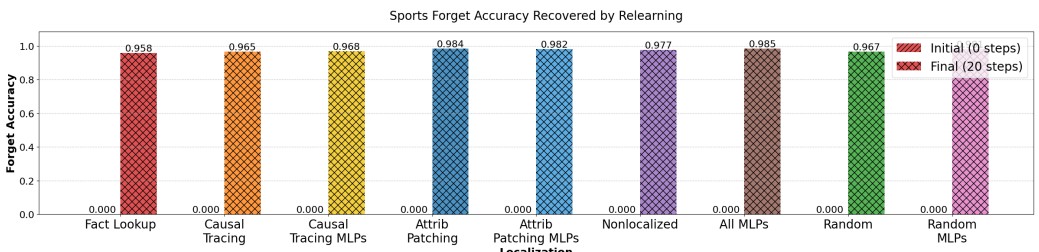

Figure 35: Relearning results for Full-Sports-Editing.

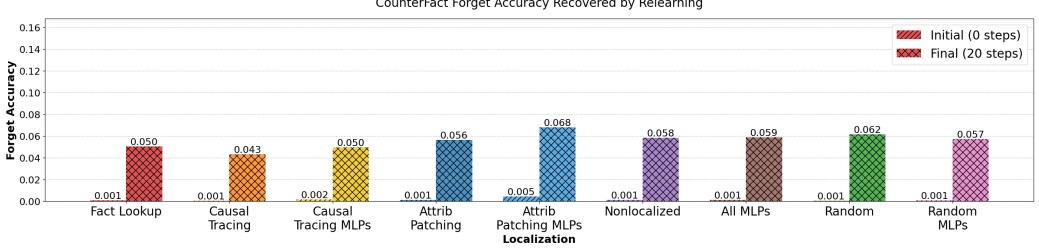

Figure 36: Relearning results for CounterFact-Editing.

four soft prompts. Soft prompt evaluations can be considered to be a more narrow form of few-shot finetuning, that is closer to searching for prompts that recover the model's knowledge.

We find limited soft prompt success: across most tasks and models, we don't recover much held-out forget set accuracy. On the Sports-Athlete-Editing, CounterFact-Editing, and Sequential-CounterFact-Editing tasks, Figures 42, 44 and 45 show that all localizations don't significantly improve in Forget Accuracy over random chance, or are about equal between localizations, after soft prompts are applied. In Figure 43, specifically for Gemma-2 on Sports-Athlete-Editing we see some

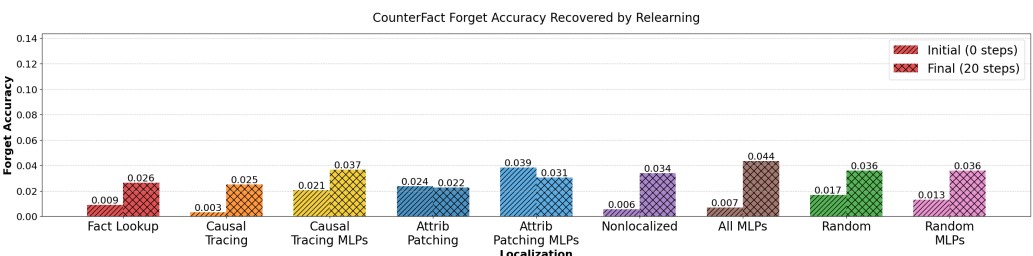

Figure 37: Relearning results for Sequential-CounterFact-Editing.

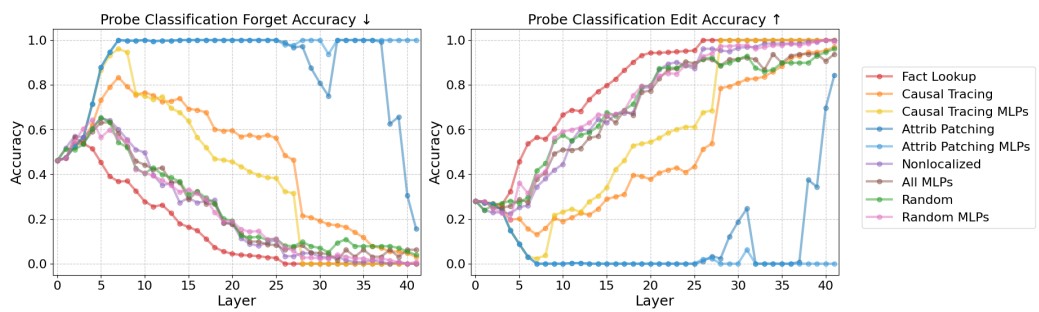

Figure 38: Linear probes applied to the forget set across all models, classifying model activations after various layers.

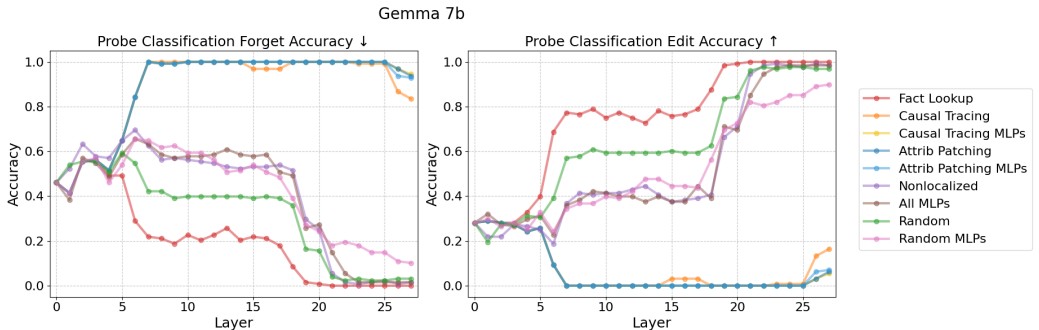

Figure 39: Linear probes applied to the forget set on Gemma-7B with 28 layers.

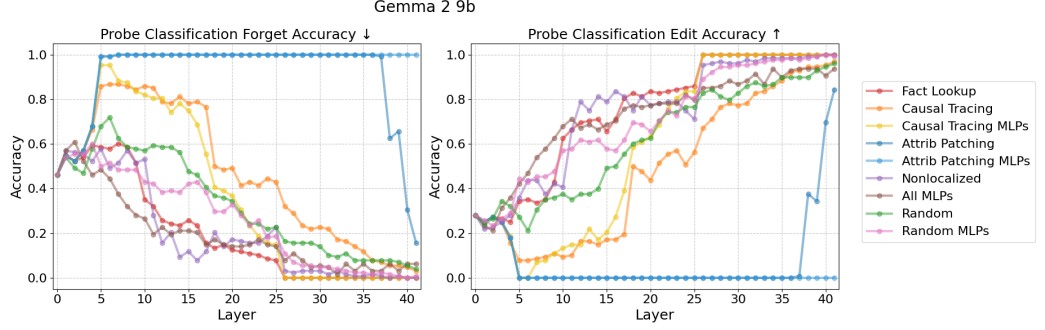

Figure 40: Linear probes applied to the forget set on Gemma-2-9b with 42 layers.

reasonable results with softprompts that are able to recover over 60% Forget Accuracy on OT localizations, while FLU, Nonlocalized, and All MLPs remain under 40% Forget Accuracy.

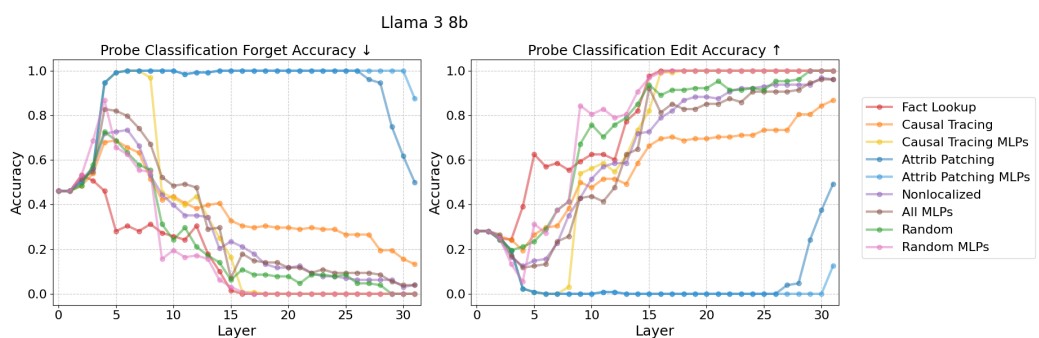

Figure 41: Linear probes applied to the forget set on Llama-3-8b with 32 layers.

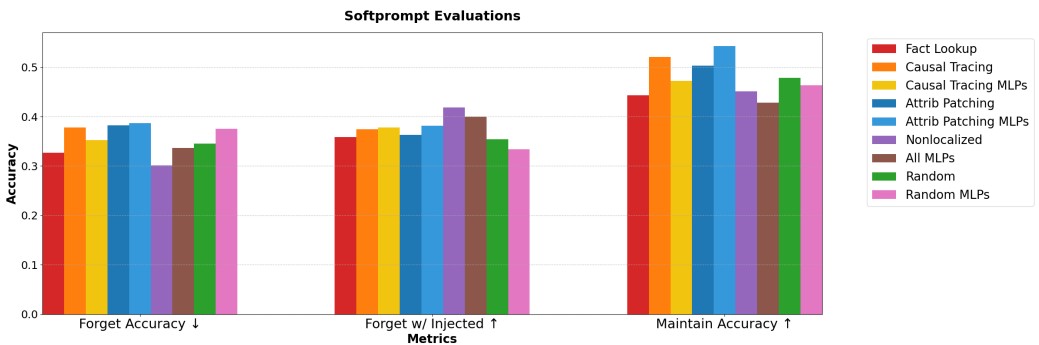

Figure 42: Metrics with soft prompts applied for Sports-Athlete-Editing, averaged over all models.

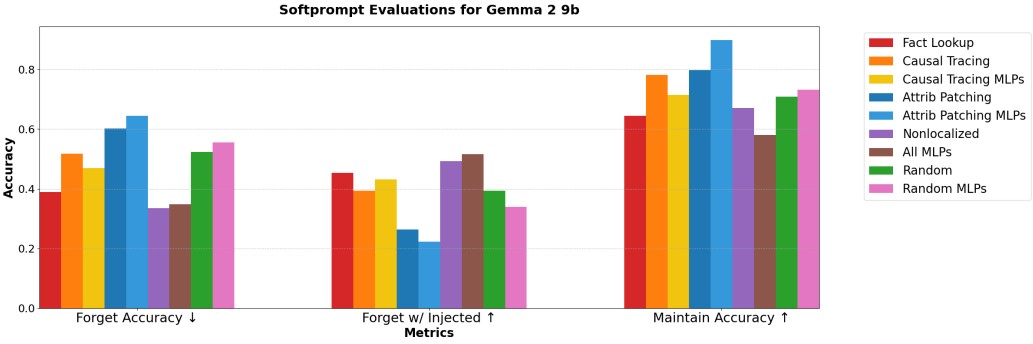

Figure 43: Metrics with soft prompts applied for Sports-Athlete-Editing for Gemma-2-9b.

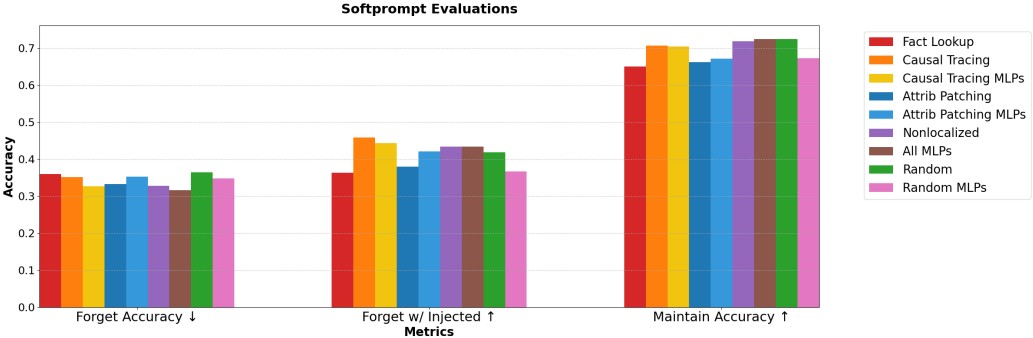

Figure 44: Metrics with soft prompts applied for CounterFact-Editing, averaged over all models.

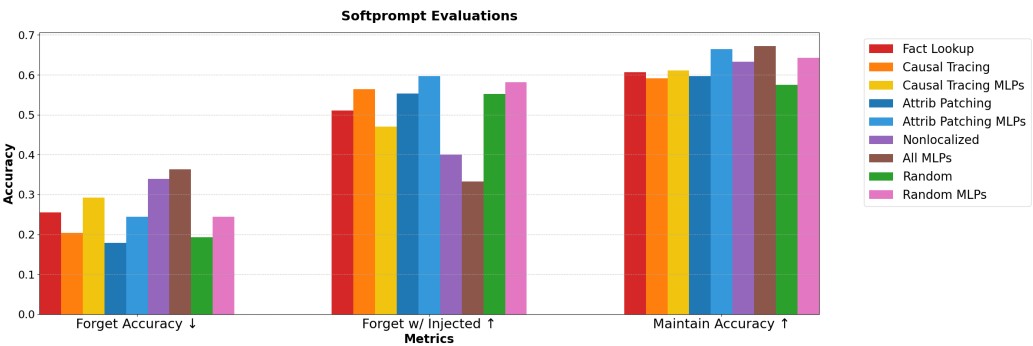

Figure 45: Metrics with soft prompts applied for Sequential-CounterFact-Editing, averaged over all models.

