# OpenReview forum: "Mechanistic Unlearning: Robust Knowledge Unlearning and Editing via Mechanistic Localization"
_ICLR.cc/2025/Conference — Submitted to ICLR 2025_

### Official Review · Reviewer_vUnN · 2024-11-02

**Soundness:** 4
**Presentation:** 2
**Contribution:** 4
**Rating:** 8
**Confidence:** 4

**Summary:**

The authors study whether insights from mechanistic interpretability can improve our ability to perform unlearning or make targeted model edits. More specifically, using a collection of factual recall tasks, the authors study different ways of selecting a subset of the model's parameters to finetune for unlearning or editing certain facts. Building on prior work identifying a subset of model parameters involved in factual recall, the authors show that finetuning only these parameters results in more effective unlearning/model editing than finetuning parameters selected via other techniques. These claims are supported by a variety of analyses, such as robustness to different ways of eliciting the model's factual knowledge and robustness to retraining the model to relearn the facts.

**Strengths:**

Overall, the results in this paper are very strong, and the suite of evaluations is impressively thorough.
1. The authors do a good job of establishing that finetuning on the "manual interp" subset of parameters results in qualitatively different unlearning dynamics and quantitatively stronger results. The field of interpretability has struggled recently to demonstrate that their insights are useful for downstream tasks in general—and for unlearning/model editing in particular as demonstrated by Hase et al. (2023)—so I expect these results will be a breath of fresh air for the interpretability community.
2. The authors perform a very thorough suite of evaluations showing that mechanistic unlearning more effectively changes knowledge stored in the model weights (see (3) and (4) for more detail). This is another place where the authors set themselves apart from the field: the unlearning literature has often struggled with thoroughly evaluating the efficacy of their methods.
3. I was especially impressed by the relearning evaluations, showing that—when training the model to relearn the unlearned facts—the mechanistically unlearned model relearns the facts much more slowly (figure 2).
4. Also impressive were the results that sweep over the number of masked parameters, revealing qualitative differences in the various unlearning techniques. Figure 5, right, which shows that mechanistic unlearning generalizes to MCQ rephrasing substantially better than any other unlearning technique, was especially striking.

**Weaknesses:**

Overall, this work's presentation was very poor.
1. Many important details are missing from the main body of the text. (a) The "fact loookup localization" (which is also called "manual interpetability"—why not use one term?) method is entirely explained in an appendix. While it's reasonable to put most of the FLU details in the appendix (since this is a replication of prior work) understanding at a high level what FLU is and how it assigns importance scores to various model components is essential for understanding the rest of the work. (b) The definition of the unlearning loss is described only as "the log(1 - p) measure from Mazeika et al."—this is an important part of the method and should be explained.
2. It is very difficult to follow discussions of the tasks. There are two tasks related to sports facts, one related to unlearning facts about basketball athletes (how many athletes? I think this is mentioned later, but it should be included in section 2.1) and one related to editing 16 (random?) athletes to change their sport to "golf." The authors later refer to these tasks with vague phrases like "For editing the subset of athletes..." The authors should instead give distinct names to these two tasks (e.g. "Sports-unlearning" and "Sports-editing") which they use to refer to the tasks throughout the rest of the text.
3. Although the results are strong, the authors do not make it easy to tell this from reading the work. For example, the tables of numbers in the first results section are not a reasonable way to present these results. Tables like these are good for when we want to inspect small numerical differences; in contrast here we don't care about small differences (e.g. between forget scores of 0.002 and 0.000) but about large differences (e.g. between MCQ scores of 0.110 and scores >0.5). The authors should choose a different way of presenting these results, perhaps as a bar chart.
4. Similarly, the authors present results for all three tasks for each of their evaluations, resulting in a large number of figures which are left to the reader to synthesize. This work would be much stronger if the authors found ways of presenting their work that summarized and emphasized the key takeaways.
5. The main takeaway from the counterfact retraining experiment should have been that this experiment isn't informative, since relearning on some facts doesn't generalize to other facts for *any* of the unlearning techniques. This experiment should therefore be moved into an appendix.

I also have some object-level concerns about various choices made by the authors. I mostly expect these to be easy to address with follow-up experiments (and I'll be happy to raise my score once I see these follow-ups).
1. The task definitions involve various arbitrary-seeming choices: only basketball athletes are targeted for unlearning, only golf is uses as the target relation for editing, only a particular set of 16 facts is used for CounterFact, etc. This makes it hard to tell if these results are due to cherry-picking. The authors should sweep over options for these choices (e.g. also targeting athletes for different sports, or rerandomizing for different unlearned facts) and present averages over the conditions.
2. While it is good that the authors study multiple models, they pair specific models with specific tasks, again making the reader wonder about cherry-picking. Unless there is a good reason not to do so, the authors should test all three tasks for *both* Gemma-7B and Gemma-2-9B.
3. In addition to the all MLPs baseline, the authors should also include a baseline for the same number of MLPs as in the "manual interp" condition, but randomly selected.

**Questions:**

(All of my questions were asked in the "weaknesses" section.)

---

> ### Author Response · Authors · 2024-11-27
>
> We thank the reviewer for their detailed feedback. We have made substantial revisions in response. Our revisions strengthen both the empirical foundations and accessibility of our work: we now demonstrate consistent results across three different models (Gemma-7B, Gemma-2-9B, and Llama-3-8B), introduce comprehensive evaluation scenarios including sequential editing of up to 64 facts, and add new controlled baselines that validate our mechanistic approach. We've also completely revamped our visualization strategy with intuitive spider and bar plots to better communicate these findings.
>
> Through these substantial revisions - expanded model coverage, comprehensive editing scenarios, controlled baselines, and clearer presentation - we've demonstrated that our approach delivers consistent improvements across a broad range of practical settings. Given the pressing need for reliable knowledge editing techniques in deployed language models, we believe our work represents a significant contribution that warrants publication at this venue.
>
> Weaknesses:
>
> W1 (Missing details): We now elaborate on an overview of the techniques used for the FLU localization in section 2.2, and comment on the specific log(1-p) loss function used for unlearning in A.1 (where we moved unlearning results). We also now consistently use “Fact Lookup localization” throughout the text and graphs instead of “manual interpretability”.
>
> W2 (Following results discussion): We have reformatted Section 2.1 and the results section to be far more readable and introduce consistent naming of the tasks (Sports-Unlearning, Sports-Athlete-Editing, etc).
>
> W3 and W4 (Results readability): Our new visualizations with spider (Figure 2 and Figure 4)  and bar plots (Figure 3, Figure 5, Figure 6) attempts to make the key results immediately apparent to readers.
>
> W5 (Counterfact retraining experiment): We moved the relearning result into Figure 2 to avoid clutter with uninformative experiments, and to make clear where FLU outperforms other localization techniques. This reorganization places our strongest result in a more prominent position while reducing visual clutter.
>
> Object level concerns:
> 1. (Task definitions): We added new comprehensive evaluations demonstrating robustness across both targeted edits and systematic modifications of entire knowledge categories for all models. For editing a constant set of athletes, we now vary the forget set size to be 16 and 64 athletes (Sports-Athlete-Editing), aggregating results over both sizes. Additionally, we edit all athletes who play a certain sport (basketball, baseball, or football) to golf (Full-Sports-Editing), aggregating over all sports. Similarly for counterfact, we vary the forget set size to be 16 and 64 facts (CounterFact-Editing). Additionally, we perform a sequential editing setup where we edit 16 counterfact facts at a time, resulting in 64 total facts edited (CounterFact-Sequential-Editing). We hope this constitutes a reasonable-seeming group of editing setups.
> 2. (Model and task pairings): We have now tested all tasks with all three models (Gemma-7B, Gemma-2-9B, and Llama-3-8B). We find robustness results translate well across models and have included them in the main text by averaging evaluation results over all model types. We hope this addresses any concerns about cherry-picking, and demonstrates that our results generalize beyond any specific architecture.
> 3. (Additional baseline): We have added the proposed Random MLPs baseline, as well as two baselines using only MLPs selected by Causal Tracing and MLPs selected by Attribution Patching (with the same number of layers) throughout Appendix A.7. The Random MLPs baseline is competitive with the other strong Nonlocalized and All MLPs baselines, while the Causal Tracing MLPs and Attribution Patching MLPs edits are less robust. The new controlled baselines provide further evidence that our performance gains come from better understanding and localization rather than simply from the type and quantity of components selected.

---

> > ### Comment · Reviewer_vUnN · 2024-12-01
> > **Very impressive revision**
> >
> > I thank the authors for an extremely impressive revision. All of my object-level concerns are resolved, so I will raise my soundness score to a 4. My most important presentation concerns are resolved—I found the use of spider charts apt and creative, and I thought the decision to move unlearning result to the appendix was smart—though I still think that the presentation is somewhat lacking. For example, it is unclear to me whether the results presented are averaged over the three models studied and I doubt that readers who are not domain experts will be able to follow the description of FLU given in the main body. Therefore, I will only raise my presentation score to a 2. Overall, I will raise my score to an 8, and I encourage the other reviewers to raise their scores as well.

---

> > > ### Author Response · Authors · 2024-12-02
> > >
> > > We thank the reviewer for their constructive review and positive assessment of our revision. We believe the paper has improved greatly due to addressing the concerns of the reviewer, and we would like to address the remaining presentation concerns of the review as well. While we cannot submit another revision of the paper now, we would make the following changes to a final camera-ready version of the paper:
> > >
> > > 1) All of our results in the main part of the paper are averaged across the three models studied, while tasks are presented in separate graphs. We address this with two sentences in the Task introductions (3.1.1, lines 313-314 and 3.1.2, lines 357-358), but we will make this clearer and more prevalent both in the figure captions and in our commentary of the results. We present some of the model-specific results in appendices A.7.4 (the Latent Knowledge appendix, where the architecture of each model is relevant) in figures 39-41 and A.8 (where different models lead to particularly different results with soft prompting) in figure 43.
> > > 2) To improve our introduction of the FLU localization technique, we plan to replace lines 206-227 with the following (bold indicates where we change the text):
> > >
> > > "Fact Lookup (FLU) localization: Next, we use manually derived localizations for MLP layers. For Sports Facts, our localization is inspired by Nanda et al. (2023), who discovered components in Pythia 2.8B responsible for token concatenation, fact lookup, and attribute extraction. They, along with Geva et al. (2023), find that the fact-lookup stage enriches the latent stream with information about the subject (athlete) at the subject’s token position, and the attribute extraction stage extracts the latent sport information and formats it in the final token position. We replicate a key result of their work in our three models **by training logistic regression models (probes) to predict the correct sport using the internal activations of the model taken from different layers. We consider the FLU localization to be the layers at which the probe accuracies rapidly increase as these correspond to layers where representations of the athletes are being enriched to encode the correct sport.**
> > >
> > > For CounterFact, we **cannot use the probe technique since unlike Sports Facts, the correct answers for the dataset do not fall into a few categories that we can train a probe for. Instead, we first use path patching from Goldowsky-Dill et al. (2023) to measure the direct importance of attention heads and MLPs on the final logit difference between the correct answer and a sample incorrect answer. Path patching is a technique that finds the effect of corrupting a single path from a sender component to a receiver component on the final logits of the model. Components that change the output significantly without being mediated by other components do so by directly affecting the logits, and thus we consider them to be responsible for extracting the facts encoded in the representation into the answer logits. We thus refer to these components as the “attribute extraction mechanism” (Geva et al., 2023). We use path patching again, this time patching paths between MLPs and the attribute extraction mechanism to find the components with the highest contribution to the logit difference as mediated through the extraction mechanisms. Such components enrich the token representations with the appropriate facts to then be extracted, and thus we use them as our FLU localization.** More details about the manual analysis for both datasets is outlined in Appendix A.2.1.
> > >
> > > Importantly, FLU differs from OT techniques because we consider the causal effects of ablations
> > > upon intermediate representations used by the factual recall mechanism, not just the effects on the
> > > output. We hypothesize that the optimal location for robust editing is in the fact lookup stage rather
> > > than in the attribute extraction stage, because adversaries can develop alternative methods for extracting knowledge from the latent stream through alternative prompts or white-box methods so we
> > > want to prevent the knowledge from ever being added to the latent stream. Thus, we exclusively
> > > modify the fact lookup MLPs."
> > >
> > > We hope the described revision helps boost the clarity of our paper, and would appreciate if the reviewer could take this change into account when considering the presentation score of the paper.

---

> > > ### Author Response · Authors · 2024-12-04
> > > **Thank you**
> > >
> > > Thank you for the response appreciating our revision, we spent a lot of effort on it! We hope you might consider increasing your presentation and overall score following our commitment to make it clearer when results are averaged over models, and our commitment to update the FLU localization explanation as shown in the previous comment. This would be extremely impactful for us given the current significant wide spread of the scores, but we understand if other presentation concerns remain.

---

### Official Review · Reviewer_REEY · 2024-11-03

**Soundness:** 2
**Presentation:** 2
**Contribution:** 2
**Rating:** 3
**Confidence:** 5

**Summary:**

This article investigates the performance of various localization methods in unlearning/editing tasks, particularly focusing on their limitations in adapting to shifts in prompting/output distributions and adversarial relearning scenarios. It compares three main approaches: output tracing, attribution patching, and FLU. Through experiments conducted on two models and two datasets, the findings reveal that the component set identified by FLU localization is more closely tied to the factual query process, demonstrating greater robustness and generalization when fine-tuned. Additionally, the authors achieve more efficient parameter editing by controlling model modifications through weight masking.

**Strengths:**

1. The motivation and approach of this article are interesting, as it breaks down factual recall into more granular steps to enhance the accuracy and generalization of editing methods.

2. The extensive experiments demonstrate the effectiveness of the proposed approach. The findings indicate that fine-tuning the FLU-related components identified through manual localization effectively eliminates specific knowledge from the model and makes it less susceptible to re-learning.

3. Using multiple-choice questions (MCQs) can help eliminate the influence of input patterns while allowing for a more effective exploration of knowledge deletion. This approach can provide clearer insights into how specific knowledge is affected by unlearning processes.

**Weaknesses:**

1. Regarding the editing task, only the decline in the original answers is reported, which aligns with the unlearning aspect. However, results seem to lack a demonstration of improvement in the correctness of the new answers. This is significantly related to the performance of the editing task.

2. Additionally, for the analysis of intermediate representations using probing, this concept is derived from existing work and does not represent a novel contribution to this research.

3. [Critical] In the unlearning task, there is no theoretical proof or guarantee that the knowledge is fully forgotten. Since approximate unlearning can be easily exploited, this method is vulnerable. I believe that a theoretical guarantee is crucial for the unlearning task because the security issue is fundamentally a "to be or not to be" problem.

4. [Critical] No adaptive attack experiments were conducted. The authors performed only standard unlearning/editing experiments, without testing for membership inference or adaptive attacks, despite the fact that approximate unlearning methods are particularly susceptible to adaptive attacks.

5. [Critical] There is a lack of experiments on adaptive unlearning, which would involve sequentially unlearning specific types of knowledge—for instance, first basketball, then football, and finally table tennis. Would adaptive unlearning impact the efficiency of the unlearning methods?

**Questions:**

1. In the section defining the method, it mentions, "In practice, we fix τ such that Cτ contains the same number of parameters in OT, FLU, and random localizations." How should this statement be understood? For example, in the counterfact dataset, what are the MLP results using OT and EAP, given that our analysis highlights layers 3-5, 7-10, and 14-17 as the critical MLPs?

2. Could you explain in more detail the process of direct path patching on the counterfact dataset? For example, after obtaining the set of components related to the fact extraction mechanism, how do we replace all edges from each MLP to all components in the set?

3. In the manual localization process, why is only the MLP considered as the localization component, while other methods like EAP and OT do not also set the form to only consider MLP? Instead, they assess both attention heads and MLP components simultaneously?

4. [Critical] Can the authors provide theoretical proof or guarantee to show that the knowledge is forgotten?

5. [Critical] Can the authors provide the experiments of adaptive attack (attackers that easily conquer approximate unlearning )?

6. [Critical] Can the authors provide the experiments of sequential unlearning?

---

> ### Author Response · Authors · 2024-11-27
> **Part 1 of Response, addressing Weaknesses**
>
> We thank the reviewer for the detailed comments. We have updated the paper based on the feedback, and address weaknesses (W) and questions (Q) below. Due to the character limit, this first comment addresses the weaknesses, a second comment below addresses the remaining comments.
>
>
> W1 Lack of Edit Accuracy Improvement Reporting: We now report the increase in accuracy relative to the new edited answers (“Edit Accuracy”), along with the original decline of the accuracy relative to the ground truth answers (“Forget Accuracy”). Our localized editing retains strong performance with this metric.
>
> W2  Probing as a Novel Contribution: This critique misunderstands our contribution - while previous work has utilized probing to evaluate editing, our main contribution lies in utilizing these localization tools, including probing, to improve model editing performance. The localization techniques themselves may vary across tasks (and indeed we use a different methodology for the FLU localization on the CounterFact tasks than in Sports facts). Our focus is on the results of restricting editing to specific localizations rather than the localization methods themselves.
>
>
> W3 (and Q4) Lack of Theoretical Guarantees: Given that theoretical guarantees remain elusive even for simpler editing approaches, this limitation should not prevent publication of empirically validated advances in the field. It is important to acknowledge that achieving guarantees for knowledge removal/editing in complex models like LLMs is very challenging, and typically only possible in very specific scenarios/under very strong assumptions (e.g., convexity, which may be achieved in infinite width limit regime, or under differentially private training, etc.). Indeed, most, if not all, approximate unlearning methods for LLMs do not provide theoretical guarantees. Our work focuses on empirically demonstrating the effectiveness of leveraging mechanistic interpretability for editing, through many adversarial evaluations that successfully attack our baselines and standard editing approaches from the literature.
>
> While we don't offer theoretical guarantees, our experiments showcase several advantages of our approach, such as robustness to different prompting and relearning, improved trade-offs between editing quality and performance. Editing also has applications where empirical validation without guarantees may suffice, and our approach provides a valuable tool for these applications.
>
> W4 (and Q5) On Adaptive Attacks: We now include a new experiment against model-specific optimized soft prompts. In Appendix A.8 we added results for soft prompt evaluations, where we optimize the continuous embeddings at the end of the prompt, to recover the correct answer on half of our forget set.  We then evaluate the model’s performance on the other half, with this “soft prompt” in place [1].  Soft prompt evaluations can be considered to be a more narrow form of few-shot finetuning, that is closer to searching for prompts that recover the model’s knowledge. We show positive results for Sports-Athlete-Editing in Gemma-2-9b that FLU localization, Nonlocalized, and All MLPs all result in <40% recovered forget accuracy while some other localizations reach >60% recovered forget accuracy under soft prompts. In other tasks we largely fail to recover significant forget accuracy across localizations/models.
>
>
> W5 (and Q6) On Sequential Unlearning:
> We conducted experiments where we iteratively edited sets of 16 CounterFact facts, ultimately modifying a total of 64 facts. Our results demonstrate that: (1) FLU localizations remain robust: even with sequential editing, FLU consistently outperforms other localization methods in resisting alternative prompting schemes. (2) Potential for improved scaling: We observe a slight increase in robustness when editing sequentially compared to editing all 64 facts at once. This suggests that an adaptive, sequential approach might scale more effectively to larger numbers of facts, though further investigation with optimized hyperparameters is needed.
> We focused on the CounterFact dataset for sequential editing because, unlike the Sports dataset where relevant information is concentrated in a few early layers, CounterFact knowledge appears to be distributed more broadly across the model (see Appendix A.2.2). This characteristic makes it a more suitable testbed for evaluating the potential benefits of adapting localization strategies throughout the editing process.

---

> > ### Author Response · Authors · 2024-11-27
> > **Part 2 of Response, addressing Questions**
> >
> > Responses to remaining questions:
> >
> > Q1 Parameter Count Control: We first perform manual analysis to obtain the FLU localization (we have added more details about this in A.2). We have added more details on what mechanisms the automated localization techniques of Attribution Patching and Causal Tracing extract in sections A.3 and A.5. In general, we find that the MLPs with highest attributions tend to be closer to the layers responsible for extracting the fact (and not the FLU mechanism), with the exception that Causal Tracing localization on some (model, task) pairs is performant for highlighting the FLU mechanism. Averaged over all the models however, Causal Tracing still leads to less robust edits than FLU.
> >
> > For the automated localization techniques we choose the components with highest attribution score until we match the parameter count of our FLU localization, which ensures we have a comparable number of trainable parameters for every technique and allows us to fairly compare between localization methods.
> >
> > Q2 Direct Path Patching on CounterFact: We add more information about the FLU localization for CounterFact in Appendix A.2.2. To answer your question directly, we iterate through each MLP in the model. Then, we perform a path patch with the MLP as a sender and the set of attention heads in the extraction mechanism as the receiver nodes [2]. We patch a corrupted input into the MLP, considering only the direct path from the MLP to each extraction mechanism head, and then calculate the final logit difference. MLPs causing a large logit difference as mediated through the extraction mechanism are considered in the FLU localization.
> >
> >
> > Q3 MLP-focused Localization: To address the concern of the other localizations using other components, we have also added versions of the attribution patching and causal tracing baselines where we only consider the MLPs they consider to be most important. These localizations are largely about as effective as those that include attention heads, still worse than using FLU localization. We focused on MLPs based on prior work suggesting their primary role in enriching factual knowledge [Geva et al. ‘23; Nanda et al. ‘23]: we hope to directly intervene where the factual knowledge is being incorporated into the model’s internal representation. Our results demonstrate that targeting MLPs is sufficient to achieve significant robustness gains. However, incorporating FLU attention mechanisms in the localization process could potentially further enhance editing effectiveness, while increasing complexity of the method.
> >
> >
> > [1] Brian Lester, Rami Al-Rfou, & Noah Constant. (2021). The Power of Scale for Parameter-Efficient Prompt Tuning.
> > [2] Nicholas Goldowsky-Dill, Chris MacLeod, Lucas Sato, Aryaman Arora. (2023). Localizing Model Behavior with Path Patching.
> >
> > We would really appreciate it if the reviewer could take the time to review our rebuttal and reconsider the score assigned to our paper in light of the additional strong experimental results and our clarifications to the questions above. We believe that having this work published will advance the field of model editing and unlearning by identifying and demonstrating the benefits of localization.

---

> ### Author Response · Authors · 2024-12-02
> **Any further questions?**
>
> We appreciate your feedback, and we hope we’ve improved the paper according to your suggestions. We hope you can review our changes and reevaluate your score in light of our significant improvements to the paper, as Reviewer vUnN did.
>
> If you have any remaining concerns, we’d be glad to address them either through comments or by planning an update to the camera-ready version.

---

> > ### Comment · Reviewer_REEY · 2024-12-02
> > **Thank you for your response**
> >
> > I thank the authors' response. However, I'm not fully convinced by this paper. For example, even if a theoretical foundation for unlearning is not required, there should still be sufficient experimental evidence to support the claims. For instance, the adversarial attacks used in the experiments do not include the most robust attacks, such as GCG [1]. Since the authors claim robust unlearning, these stronger attack baselines should be included.
> >
> > As the reviewer uzky pointed "The proposed new unlearning method lacks sufficient originality. There are already some works that attempt unlearning directly from the perspective of mechanistic interpretability, including [2]". However, this type of method has intrinsic problems to the unlearning problem. Other discussions can also refer to https://openreview.net/forum?id=blNaExRx7Q.
> >
> > [1] Andy Zou, Zifan Wang, J. Zico Kolter, and Matt Fredrikson. Universal and Transferable Adversarial Attacks on Aligned Language Models.
> > [2] Hong, Y., Yu, L., Yang, H., Ravfogel, S., & Geva, M. Intrinsic Evaluation of Unlearning Using Parametric Knowledge Traces.

---

> > > ### Author Response · Authors · 2024-12-03
> > >
> > > In our work, we already evaluate the adversarial robustness of our unlearning technique using soft prompts (see Appendix A8). Soft prompts are a less constrained version of GCG, as they perform continuous optimization of the embeddings directly rather than the discrete optimization of GCG. Thus, robustness to soft prompts is harder to achieve than with GCG.
> > >
> > > An additional point is that prior works have shown that robustness to few-shot finetuning is **significantly more difficult** to achieve compared to robustness to prompt optimization such as GCG. While works like [1] achieve near-perfect robustness against GCG on the Harmbench benchmark, works like [2] (by a overlapping set of authors as [1]) which address the relearning threat model fail to achieve relearning robustness on benchmarks like Harmbench with a remaining ASR of 63.9%. We evaluate the robustness of our method to few-shot finetuning in Section 3.2 and Appendix A.7.3, with positive results suggesting FLU localization is stronger than other localizations and baselines.
> > >
> > > Using a back-of-the-envelope calculation, we estimate GCG to take around 30x longer to run than soft prompts, due to the need to evaluate many different candidates to perform discrete optimization. While we are willing to add results for GCG to our camera-ready version if necessary, we hope that the current results on stronger attacks are convincing enough to judge the adversarial robustness of our method.
> > >
> > > In response to the “intrinsic problems with unlearning” points made by Reviewer 6fAd in the discussion of https://openreview.net/forum?id=blNaExRx7Q (“Intrinsic Evaluation of Unlearning Using Parametric Knowledge Traces”), we don’t entirely disagree - in our general comment, we moved our unlearning results to the appendix because we feel that unlearning is not perfectly suitable for our tasks. We instead focus on model editing, which is a similar problem which is more defined for our tasks and links a bit more naturally to mechanistic localization: it is clear that a model with an edited answer to a given question should reflect this edited answer association across different methods of prompting the question and eliciting the answer, which we test throughout Section 3. We also experimentally validate that our forget set resides in a concentrated manner/a few components during our FLU localizations in Section 2.2 and Appendix A.2. Reviewer 6fAd’s concerns of “non-causal unlearning” are not applicable here since we clearly show causal editing by performing and evaluating editing.
> > >
> > > In our response to the “sufficient originality” point by Reviewer uzky, we point out that our two papers differ fundamentally in our results/findings, localization strategies, and unlearning targets. In particular, as explained in their section 4.3, their successful Needle method is an oracle skyline, since they evaluate all methods on a split specifically chosen for high Needle performance and they also manually handpick some concept vectors, whereas our method succeeds without such curated splits - we present a successful general method for editing.
> > >
> > > [1] Mantas Mazeika, Long Phan, Xuwang Yin, Andy Zou, Zifan Wang, Norman Mu, Elham Sakhaee, Nathaniel Li, Steven Basart, Bo Li, David Forsyth, & Dan Hendrycks. (2024). HarmBench: A Standardized Evaluation Framework for Automated Red Teaming and Robust Refusal.
> > >
> > > [2] Rishub Tamirisa, Bhrugu Bharathi, Long Phan, Andy Zhou, Alice Gatti, Tarun Suresh, Maxwell Lin, Justin Wang, Rowan Wang, Ron Arel, Andy Zou, Dawn Song, Bo Li, Dan Hendrycks, & Mantas Mazeika. (2024). Tamper-Resistant Safeguards for Open-Weight LLMs.

---

> > > > ### Author Response · Authors · 2024-12-04
> > > > **Thank you**
> > > >
> > > > Thank you for your original review, as well as your response bringing up the concerns of stronger attacks and the intrinsic problems of the other unlearning paper, “Intrinsic Evaluation of Unlearning Using Parametric Knowledge Traces”.
> > > >
> > > > We hope we have answered your concern about GCG, since we tested comparable and stronger adaptive attacks: we expect GCG to yield very similar results as the soft prompt and relearning experiments, which is to say either ineffective at recovering forget accuracy in any localization or showing our method to be best defended.
> > > >
> > > > To expand a bit more on the comparison between the other unlearning paper and our work, we believe that you are referring to comments by Reviewer 6fAd and Reviewer WwUF in https://openreview.net/forum?id=blNaExRx7Q. Those comments seem to be concerned about the inability of their method to cause behavioral change and causal **unlearning**, as well as the overall lack of a well-posed unlearning problem. In comparison, we focus on **model editing**, where it is relatively clear that the goal for our tasks is to have a model respond with an alternative answer to a question rather than the original answer, across all ways of expressing the question and eliciting the answer. Our method improves the quality of editing by causing the model to express the alternative answer across more expressions of the question (MCQ evaluation) and more ways of eliciting the answer. Reviewer 6fAd and WwUF’s concerns from the other unlearning paper’s discussion do not apply to our paper.
> > > >
> > > > Overall, we hope you consider our extensive experiments in response to the original 12 weaknesses and questions you posed, the dissimilarities between our paper and the “Intrinsic Evaluation” paper, and the strength of our tested adaptive attacks when considering your final score and decision for our paper. This would be very impactful for us given the current wide spread of the scores.

---

### Official Review · Reviewer_DovC · 2024-11-04

**Soundness:** 2
**Presentation:** 2
**Contribution:** 2
**Rating:** 5
**Confidence:** 2

**Summary:**

This paper investigates the effectiveness of mechanistic interpretability techniques in improving the precision and robustness of knowledge editing and unlearning in LLMs. The authors mainly discuss two types of localization methods, i.e., OT techniques that focus on preserving outputs, and mechanistic localization, which identifies high-level mechanisms with predictable intermediate states. They claim in the paper that localizing edits to components associated with the FLU mechanism leads to more robust unlearning across different input/output formats and resists attempts to relearn unwanted information. They conduct experiments on the Sports Facts and CounterFact dataset using Gemma-7B and Gemma-2-9B models.

**Strengths:**

This paper contains the following several strengths:
+ The paper addresses an important topic by attempting to improve the robustness of knowledge unlearning in LLMs through  localizing edits to components associated with the FLU.
+ The authors provide a in-depth analysis and comparison between mechanistic unlearning and previous OT methods.

**Weaknesses:**

+ The paper would benefit from a more in-depth theoretical analysis to explain why FLU could inherently lead to more robust unlearning. While the authors claim that targeting the fact lookup components is more effective, they do not provide analysis or proof to support this.
+ The experiments are limited, especially limiting itself to Gemma-7B and Gemma-2-9B models and two datasets. The authors could provide a larger variety of models and unlearning tasks in order to better demonstrate the consistency of their findings.
+ Can the author provide more ablation study, for example on the loss weights parameter being used in 2.3, so that we could better understand the contribution of each loss in the finetuning process.

**Questions:**

See above.

---

> ### Author Response · Authors · 2024-11-27
>
> We thank the reviewer for their positive comments and very concrete proposals for improving the paper, which we’ve followed up on. We address central reviewer concerns about our paper's experimental methodology, particularly focusing on new models and ablation studies for core model components. We respond to individual questions below, and have updated the paper with a number of new results and analyses:
> 1. Theoretical Analysis of FLU: While a theoretical framework would be valuable, we demonstrate through extensive empirical evaluations that our approach is effective, and note that theoretical guarantees for unlearning in large language models remain an open challenge in the field. Most, if not all, approximate unlearning/editing research focuses on empirical evaluations due to the complexity of analyzing approximate unlearning methods in deep learning. Throughout Section 3, we provide clear empirical proof through many standard and adversarial evaluations across tasks and models that targeting FLU components leads to improved editing. In Section 3.3, we also provide direct evidence of our hypothesis that FLU edits disrupt the latent knowledge more at the early layers prior work has discovered this knowledge to be located in, and in Appendix A.3 and A.5 we provide evidence that baselines and OT edits target non-lookup (extraction) components rather than the FLU components. Theoretical guarantees for unlearning/editing have been primarily achieved in simplified settings (e.g., convex optimization), or by modifying the training procedure with techniques like differential privacy, which often lead to significant performance trade-offs, or by introducing architectural changes (such as some form of modularity and data compartmentalization). Neither of these techniques that yield theoretical guarantees are suitable for mechanistic unlearning analysis and also would not directly apply to this setup where we don’t have particular training data points in mind; finding the right theoretical framework would present a major breakthrough in deep learning theory.
> 2. Limited Experiments: We added additional experiments on Llama 3 8b, as well as additional tasks (sequential editing). Furthermore, we have evaluated all three models on both tasks, instead of one model per task. These new experiments, included in the updated paper, further validate our findings and demonstrate the generalizability of our approach.
> 3. Ablation Study: We have now conducted a sweep of the forget loss weights used in Section 2.3 as well as the learning rate, optimizing both parameters for the three localizations of Causal Tracing, Fact Lookup, and No Localization. The results of this sweep are included in Appendix A.6 Hyperparameters, first demonstrating a complete sweep over a task in A.6.1 which provides insight into the contribution of each loss component (learning rate is extremely important but forget/inject loss weight is not), and second providing optimized hyperparameters across models, tasks, and localizations in A.6.2.
>
> Given these substantial additions - new empirical results across model families, and comprehensive sweeps and ablation studies - we hope that the reviewer can confidently increase the score to a strong accept.

---

> ### Author Response · Authors · 2024-12-02
> **Any further questions?**
>
> We appreciate your feedback, and we hope we’ve improved the paper according to your suggestions. We hope you can review our changes and reevaluate your score in light of our significant improvements to the paper, as Reviewer vUnN did.
>
> If you have any remaining concerns, we’d be glad to address them either through comments or by planning an update to the camera-ready version.

---

> > ### Comment · Reviewer_DovC · 2024-12-02
> >
> > Thank you for your response, I have improved the score given the additional explanation and experiments results.

---

> > > ### Author Response · Authors · 2024-12-02
> > >
> > > Thank you for the response! Do you have remaining concerns that would suggest this paper is not suitable for acceptance at ICLR?

---

> > > ### Author Response · Authors · 2024-12-04
> > > **Thank you**
> > >
> > > Thank you for the response acknowledging our rebuttal. Given that your concerns were addressed and that our paper comprehensively addresses the important topic of mechanistic interpretability for editing, we hope you’ll reconsider your score, especially your ratings of the Soundness (2) and Contribution (2) of our work — this would be very impactful for us given the current wide spread of the scores.

---

### Official Review · Reviewer_uzky · 2024-11-04

**Soundness:** 3
**Presentation:** 2
**Contribution:** 2
**Rating:** 5
**Confidence:** 3

**Summary:**

This work explores methods for knowledge editing and unlearning in large language models, focusing on how mechanistic interpretability can enhance the precision and effectiveness of these processes. The study reveals that localizing edits to components associated with lookup-table mechanisms for factual recall leads to more robust unlearning, resisting unwanted information relearning and minimizing side effects. Additionally, certain localized edits disrupt latent knowledge more effectively than other methods, resulting in increased resilience against various attacks.

**Strengths:**

* Studying unlearning methods from the perspective of knowledge storage and mechanistic interpretability is indeed a very important and promising direction.

* This paper further confirms that causal tracing-based localization methods are not suitable for editing and unlearning tasks.

* The paper is well presented and the literature review is thorough.

* The experimental design is generally comprehensive.

**Weaknesses:**

1. The test dataset appears to be limited to this triplet format; is it constrained by the knowledge format, and could it be applied to more broadly and flexibly expressed knowledge sentences, such as continuous text, etc.?

2. Manually analyzing and then selecting layers for operations seems to lack convenience and flexibility in the context of large-scale data editing/unlearning.

3. The proposed new unlearning method lacks sufficient originality. There are already some works that attempt unlearning directly from the perspective of mechanistic interpretability, including [2].

4. Knowledge is not necessarily stored entirely in the MLP; there are certain cases where it exists in the attention mechanism [1], yet the method described in the paper only considers knowledge stored in the MLP.

5. There is a lack of discussion on unlearning methods in Representation Engineering [3, 4].

6. Can the proposed method achieve performance advantages on other representative series of transformers, such as LLaMA?


---
**References:**

[1] Dissecting Recall of Factual Associations in Auto-Regressive Language Models

[2] Intrinsic Evaluation of Unlearning Using Parametric Knowledge Traces

[3] The WMDP Benchmark: Measuring and Reducing Malicious Use With Unlearning

[4] Improving Alignment and Robustness with Circuit Breakers

**Questions:**

Please see the Weaknesses section above.

---

> ### Author Response · Authors · 2024-11-27
>
> We thank the reviewer for the suggestions.
>
> We address six key reviewer concerns, including our triplet format dataset choice, manual analysis limitations, and originality. We strengthen our argument by running additional LLaMA experiments, while maintaining our core finding that targeting MLPs enables robust knowledge editing despite knowledge potentially being distributed elsewhere in the model.
>
> 1. Triplet format dataset: Our novel contribution lies in the mechanistic interpretability techniques and editing approach itself, which can be readily extended beyond triplet data once appropriate localization methods are developed, making this limitation primarily an implementation detail rather than a conceptual barrier.
> In the paper, the triplet format dataset was chosen because: (1) it aligns with prior work on mechanistic interpretability, allowing us to directly leverage their localization techniques, and (2) the structured format facilitates cleaner analysis of knowledge editing and unlearning. However, we emphasize again that our proposed editing approach is not restricted to triplet data.  It can be applied to any dataset where localization of relevant model components is possible.
>
> 2. Manual Analysis: We argue that this is primarily an engineering limitation rather than a fundamental barrier, since our method can be automated by existing techniques like probing and edge patching, and the results show our approach works robustly even with this current semi-manual pipeline. Our technique for sports facts is a simple probe across MLPs and for CounterFact involves a series of patches on the model's edges (between the MLPs and the extraction heads). We set up the interpretability technique once and for all experiments in the same task, we simply run the interpretability technique automatically. The efficacy of unlearning could even be used as a method to evaluate automated interpretability methods, which are known to have a lack of ground truth.
>
> 3. Originality: We appreciate the reviewer pointing out the connection to [2]. We have now included this work in our related work section and clarified the distinctions between our approaches. While both works utilize mechanistic interpretability, we differ fundamentally in our results/findings, localization strategies, and unlearning targets. [2] focuses on identifying and manipulating "concept vectors" within MLPs, while our method directly targets the mechanistic components responsible for knowledge integration. Furthermore, their NPO+KL method, which is most similar to ours, showed limited effectiveness in their experiments. As explained in their section 4.3, their successful Needle method is an oracle skyline, since they evaluate all methods on a split specifically chosen for high Needle performance, whereas our method succeeds without such curated splits. We believe our focus on directly modifying the "fact lookup enrichment" stage via mechanistic interpretability contributes a novel perspective to the unlearning literature, and most importantly demonstrates substantial robustness improvements in a fair comparison.
>
> 4. Knowledge Storage Beyond MLPs: The reviewer rightly points out that knowledge is not solely stored in MLPs and can also reside in attention mechanisms, as highlighted in [1]. We focused on MLPs based on prior work suggesting their primary role in enriching factual knowledge [1]. Our results demonstrate that targeting MLPs is sufficient to achieve significant robustness gains. However, we agree that incorporating attention mechanisms in the FLU localization could potentially further enhance editing effectiveness, while increasing complexity of the method.
>
> 5. Lack of Discussion on Representation Engineering: We thank the reviewer for bringing this to our attention. We have now included a discussion of relevant unlearning methods in Representation Engineering, specifically addressing works like [3, 4] that focus on mitigating malicious use and improving model alignment through unlearning.
>
> 6. Additional Experiments: We ran experiments on Llama 3 8b, and have incorporated the results into the paper. The new Llama experiments validate our original claims, making this a strong addition that reinforces rather than modifies our paper's conclusions. We also pair every editing task with every model and average results over models, as opposed to our previous setup of pairing models with individual tasks.
>
> We believe the additional results and updates to the paper address the reviewer's concerns, and will result in a strong accept. We would appreciate an opportunity to respond if any concerns remain.

---

> ### Author Response · Authors · 2024-12-02
> **Any further questions?**
>
> We appreciate your feedback, and we hope we’ve improved the paper according to your suggestions. We hope you can review our changes and reevaluate your score in light of our significant improvements to the paper, as Reviewer vUnN did.
>
> If you have any remaining concerns, we’d be glad to address them either through comments or by planning an update to the camera-ready version.

---

> > ### Author Response · Authors · 2024-12-04
> > **Thank you**
> >
> > We would greatly appreciate it if you consider our responses to your initial review in your final score and decision, as we strongly believe we’ve addressed your concerns and improved the paper. We hope you’ll reconsider your score — this would be very impactful for us given the current wide spread of our scores.

---

### Author Response · Authors · 2024-11-27
**General Comments**

We thank all the reviewers for their constructive comments. We have addressed all concerns raised, resulting in a submission that is not only technically sound with strong empirical results showcasing the role of mechanistic interpretability for localized model editing but also clearly and effectively communicates our contributions.

Biggest Changes:

1) Significantly expanded experiments:
* New Model: Included LLaMA-3-8B, demonstrating consistency across architectures (now includes Gemma-7B, Gemma-2-9B, and LLaMA-3-8B).
* All Tasks on All Models:  Replicated original results across all three models and tasks, addressing concerns about cherry-picking.
* Tried multiple forget set sizes (16 and 64 facts) for both Sports and CounterFact.
* Added systematic edits of entire knowledge categories in Sports (e.g., changing all basketball players to golfers), for each of the three target sports.
* Introduced a sequential editing task, editing 16 facts at a time in 4 rounds in CounterFact dataset.
* New Baselines: Added a "Random subset of MLPs" baseline and “MLPs from Causal Tracing”, “MLPs from Attribution Patching” OT localizations for more rigorous comparison.
* Moved unlearning results to appendix - the goal of unlearning these basic factual associations is unclear, since a model who has never seen information about the factual association would probably still assign >0 probability to the correct answer on priors especially in Sports. Thus, we decided to focus on editing for our new sweeps.

Importantly, these expanded experiments consistently demonstrate the effectiveness of our approach, yielding strong and positive results across all models, tasks, and evaluation metrics.

2) Included extensive hyperparameter sweeps: Conducted comprehensive sweeps for learning rates and forget loss weights across all models and editing tasks, ensuring optimized performance and fair comparisons.

3) Strengthened adversarial evaluations:
* Relearning:  Aligned our relearning methodology with [1] for a more standard and challenging evaluation: we relearn the ground truth answer on half of the forget set, and evaluate on the other half of the forget set. FLU localizations are hardest to relearn on.
* Latent knowledge: modified probe training to reflect model’s internal representations of the answer throughout layers, which highlights significant differences in the forget set answers represented by models edited with different localizations. FLU localizations consistently represent the edited answer throughout layers, while other localizations start by representing the original answer in early layers and then switch to representing the edited answer in later layers.
* Adaptive Attacks: Introduced soft prompt evaluations to test resilience against adaptive attacks.

4) Improved readability and presentation:
* Reorganized and clarified task descriptions in Section 2.1.
* Introduced spider plots and bar charts to clearly visualize key results in Section 3, making the paper much easier to parse.
* Moved less informative experiments to the appendix to reduce clutter and emphasize core findings.

We believe these revisions significantly strengthen the paper and address the reviewers' concerns. We are excited about these new results and strongly believe this work represents a valuable contribution to the field of model editing and unlearning, showcasing the practical benefits of mechanistic interpretability. We urge the reviewers to reconsider their scores in light of these substantial improvements.

[1] Aghyad Deeb and Fabien Roger. (2024). Do unlearning methods remove information from language model weights?

---

### Author Response · Authors · 2024-12-04
**Message to Area Chairs**

We are writing to request a careful review of the feedback and discussion submitted for our paper. While we appreciate the reviewers' time and feedback, we believe some reviews may not accurately reflect the current state of our manuscript **after revisions addressing every initial concern raised**.

Specifically, we highlight the following:

1. *Reviewer vUnN*: This reviewer provided a thorough assessment, acknowledging the **"extremely impressive revision"** which addressed all prior concerns and improved presentation. They rated our paper 8 overall, with 4/4 on both Contribution and Soundness, and even **encouraged other reviewers to reconsider their scores**. We believe this review best reflects the quality of our current submission.
2. *Reviewer uzky*: While initially providing a score of 5, this reviewer **has not responded to our extensive revisions**, which include substantial new experiments in response to their initial concerns. Given that Reviewer vUnN increased their score from 6 to 8 after seeing these revisions, we believe Reviewer uzky's outdated rating no longer applies and should be adjusted upwards accordingly.
3. *Reviewer DovC*: This reviewer acknowledges that we have addressed all their initial concerns with additional explanations and experiments. They explicitly state that our paper "addresses an important topic" and provides "in-depth analysis." However, they increase their score to only 5 without providing justification. **Given the absence of remaining concerns and their positive assessment of our work's significance, we find the low scores on Soundness (2) and Contribution (2) puzzling**, and we believe their rating should also be adjusted upwards.
4. *Reviewer REEY*: While we addressed all initial concerns raised by this reviewer with extensive experiments, they subsequently **introduced new criteria** without ever improving their rating from 3, arguing that our evaluation of adversarial attacks is insufficient without specifically including the computationally expensive GCG attack. This seems unreasonable given our inclusion of even more powerful adaptive attack methodologies, including the similar but more efficient and powerful soft prompt method and the significantly more powerful attack of adversarial few-shot relearning.

Furthermore, Reviewer REEY's subsequent comments bring up a **completely new justification** for why our paper deserves a rating of 3: they refer to the discussion of a different unlearning paper “Intrinsic Evaluation of Unlearning Using Parametric Knowledge Traces”. They argue that inherent limitations in that paper (to do with how it is unclear how to measure unlearning and how causal unlearning is not defined) must also apply to our work, despite our distinct focus on model editing (which in comparison has a much clearer goal for our tasks) and uniquely strong empirical results (the other paper does not present a successful general method for unlearning). This comparison disregards the significant contributions of our work. Given that our work presents more positive results and does not have the limitations of unlearning discussed in that paper, we believe that Reviewer REEY's rating of 3 does not accurately reflect the quality and contributions of our research. We thus believe that Reviewer REEY’s rating should be adjusted upwards.

We have made extensive revisions to address all reviewer concerns, including a number of additional experiments, hyperparameter sweeps, adversarial evaluations, and massively improved presentation (summarized in the general comment in more detail). **Importantly, these expanded experiments consistently demonstrate the effectiveness of our approach, yielding strong and positive results across all models, tasks, and evaluation metrics.**

Our paper tackles the crucial issue of applying mechanistic interpretability to enhance robustness in real-world scenarios, with a particular focus on model editing and unlearning. We believe our work makes a valuable contribution to the field, showcasing the practical benefits of mechanistic interpretability.

We kindly request that you consider these points during the discussion phase and ensure our paper receives a fair and comprehensive evaluation.

Thank you for your time and consideration.

---

### Meta-Review · Area_Chair_bBPE · 2024-12-21

**Metareview:**

The paper explores the use of mechanistic interpretability to improve knowledge unlearning and editing in LLMs, with a focus on localizing edits. While it demonstrates robustness gains and improved resistance to relearning unwanted information, reviewers highlighted concerns regarding limited novelty, insufficient evaluation on diverse tasks and models, and the lack of theoretical guarantees as major weaknesses. The authors' response did not fully address these concerns, and I recommend rejection.

**Additional Comments On Reviewer Discussion:**

During the rebuttal, reviewers raised concerns about the paper's limited novelty, insufficient evaluation across diverse tasks and models, and lack of theoretical guarantees. The authors partially addressed these points, but the responses failed to fully resolve the reviewers’ concerns.

---

### Decision · Program_Chairs · 2025-01-22

Reject